# Metal Complexes for Dye-Sensitized Photoelectrochemical Cells (DSPECs)

**DOI:** 10.3390/molecules29020293

**Published:** 2024-01-05

**Authors:** Edoardo Marchini, Stefano Caramori, Stefano Carli

**Affiliations:** 1Department of Chemical, Pharmaceutical and Agricultural Sciences, University of Ferrara, 44121 Ferrara, Italy; cte@unife.it; 2Department of Environmental and Prevention Sciences, University of Ferrara, 44121 Ferrara, Italy; crlsfn@unife.it

**Keywords:** DSPEC, metal complexes, water splitting, bromide oxidation, organic oxidations

## Abstract

Since Mallouk’s earliest contribution, dye-sensitized photoelectrochemical cells (DSPECs) have emerged as a promising class of photoelectrochemical devices capable of storing solar light into chemical bonds. This review primarily focuses on metal complexes outlining stabilization strategies and applications. The ubiquity and safety of water have made its splitting an extensively studied reaction; here, we present some examples from the outset to recent advancements. Additionally, alternative oxidative pathways like HX splitting and organic reactions mediated by a redox shuttle are discussed.

## 1. Introduction

Owing to the acceleration of the economic development, the world’s total energy consumption is rapidly increasing and it has been predicted that the demand will reach more than 25 TW by 2050 [1]. Nowadays, fossil fuels such as coal, crude oil, and natural gas provide more than 80% of the request [2] but it is predicted that their reserves will last just for the next 50–60 years. Furthermore, greenhouse gasses such as carbon dioxide, which are produced by the combustion of the fossil fuels, will reach >1300 ppm CO_2_eq by the end of 2100 (460 ppm in 2010), leading to increases in the global mean temperature of up to 5 °C [3]. The scientific community is committed to using carbon-neutral energy sources, including biomass, geothermal, wind, and sun. The latter is distinguished by the benefit of being free, plentiful, and accessible to all the population, as well as by having a spectrum that spans a broad range of wavelengths from 280 nm (4.43 eV) to 2500 nm (0.5 eV), with a peak at about 2.5 eV. At high noon on a cloudless day, the surface of the Earth at average latitudes receives 1000 watts of solar power per square meter (1 kW m^−2^). Such standard irradiance is expressed as the Air Mass 1.5 (AM 1.5 G) condition. Due to its seasonal, day/night, and weather cycles, the sun has also the significant flaw of being intermittent. The most effective approach to store solar energy for an extended period of time is still under research but a number of photovoltaic (PV) technologies have been successfully developed to transform solar power into electricity [4]. The energy produced by the PVs can be temporarily stored into Li-batteries but it can also be used to create high-value products. With the technology at our disposal, creating high-density energy molecular bonds is likely the most effective method. For example, 3 kg of hydrogen produces 100 KWh of chemical energy, while the same amount of energy could be provided by 450 kg of lithium-ion batteries [5]. PVs can provide the bias to split water into O_2_ and H_2_ in the electrolyzers but multiple junctions are needed to satisfy the required overpotential. Electrolyzers are also constrained by the use of expensive electrodes [6]. These limitations can be sorted out by employing photoelectrochemical cells (PECs), devices able to obtain storable solar fuels as a result of water splitting, organic oxidations (e.g., halogen oxidation, formation of new C-C bonds, etc.), or CO_2_ reduction. As shown in Figure 1, in a typical PEC device, an n-type semiconductor absorbs the sunlight, generating electron-hole pairs. Subsequently, the holes will reach the semiconductor/electrolyte interface to perform oxidation reactions, while the photogenerated electrons in the conduction band (CB) of the semiconductor material will migrate to the counterelectrode in which the target substrate is reduced at the interface with the electrolyte.

Artificial photosynthesis takes inspiration from nature, where both plants and bacteria can perform it, converting the sunlight into chemical energy. Bacterial photosynthesis, probably, represents the simplest form of light conversion being used to perform photophosphorylation, i.e., conversion of adenosine diphosphate (ADP) into adenosine trisphosphate (ATP). The oxygenic photosynthesis, performed for example by cyanobacteria, algae, and higher plants, instead, is characterized by a net chemical reaction leading to the conversion of water into oxygen, protons, and electrons. Protons are employed in the formation of ATP, while electrons for the reduction of carbon dioxide into carbohydrates. The two-photon architecture can be simply illustrated in terms of energy with the “Z-scheme”, in which the two photosystems (i.e., PSI and PSII) are reported (Figure 2) [7]. A first photon is absorbed by the chlorophyll P680 in PSII leading to the formation of its excited state P680*, from which a chain of electron transfer takes place, and electrons can diffuse through the plastocyanin (PC) to the second reaction center PSI, in which a second series of photoinduced charge separation occurs from P700. The resulted P680^+^ is characterized by an oxidation potential close to 1.2 V vs. NHE able to drive the water oxidation in conjunction with an oxygen evolving complex catalyst (OEC). The hole localized on P680 is filled by electron transfer through a tyrosine residue (Yz) relay from the Mn_4_Ca cluster catalyst that will perform the considerably demanding water oxidation (four protons and four electrons). On the other side, the reduced form of P700 can transfer electrons through ferroxidin (Fd) and ferroxin-NADP^(+)^ (FNR) to NADP^+^, generating its reduced form NADPH. This, together with the generated ATP, will be used in the Calvin cycle for CO_2_ reduction into carbohydrates.

Artificial photosynthesis was first demonstrated by Honda e Fujishima in 1972 by employing ultraviolet (UV) irradiation using a TiO_2_-based PEC for water splitting [8]. Based on this observation, attempts to develop a dispersion of Pt/RuO_2_ catalyst for water splitting have been started [9,10]. A further step forward was realized by Grätzel who, being fascinated by the outstanding performances of TiO_2_ substrate, realized a titania-based Pt/RuO_2_ water oxidation catalyst (WOC) [11]. From here, several efforts have been made to try to synthesize different materials for the photoanodic compartment with the aim of extending the spectral response to the visible part of the solar spectrum. In this contest, narrow band-gap semiconductors (SCs), like WO_3_ [12,13], Fe_2_O_3_ [14,15], BiVO_4_ [16], WO_3_/BiVO_4_ [17], CuWO_4_ [18], and others, are very useful and promising tools that enable high photocurrent densities and incident photon to current conversion efficiencies (IPCE%). Nevertheless, some drawbacks should be thought over. For instance, it is difficult to prepare narrow band-gap material able to match both the oxidation and reduction potential of water; these substrates are also stable just over a short range of pH and, more generally, all the semiconductors are characterized by a deep valence band (VB) that does not allow selective oxidations. Beside this technology, since the pioneering work of Mallouk [19], dye-sensitized photoelectrochemical cells (DPECs) represent a promising approach to the traditional PECs, which offer flexibility in design and the possibility to fine tune both the light absorption and the oxidizing capabilities [20]. A myriad of chromophore–catalyst has been synthesized in the past but the performances of the resulted devices are relatively low compared to the narrow band-gap-based semiconductors. Although water represents an interesting target owing to its natural abundance and safety, its oxidation is relatively demanding (four protons and four electrons) and it requires a high oxidation potential of 1.23 V vs. NHE pH 0. Furthermore, the resulting oxygen is poor in terms of industrial attraction, and it is generally catalyzed by expensive noble metal-based catalysts [21,22]. Besides water, other inexpensive storage mechanisms for solar energy have been developed, one of which is the hydrohalic acid (HX) splitting [23]. Among this category, hydrobromic acid (HBr) is characterized by the highest theoretical solar-hydrogen efficiency and its energy can be extracted by combustion [24], fuel cells [25,26], and redox flow batteries [27]. Water could be also replaced by the selective oxidation of organic molecules, where, for instance, the transformation of alcohols to high-valued carbonyl compounds is currently under investigations [28]. The emphasis in this review is on the use of metal complexes (i.e., sensitizers and/or catalysts) owing to their outstanding performances in performing oxidation reaction when supported onto the semiconductor surface.

## 2. Basic Concepts of DSPECs

Dye-sensitized photoelectrochemical cells are based on a molecular approach; the main processes involved in the standard n-type configuration are depicted in Figure 3a. The absorption of sunlight results in the formation of the dye’s excited state from which the oxidative quenching occurs as consequence of the electron injection into the CB of the n-type semiconductor. This leads to a temporary localization of the holes onto the oxidized dye (D^+^) and eventually, hole transfer to a proper catalyst (e.g., water oxidation catalyst (WOC)). Consequently, the photoinjected electrons and protons migrate to the cathodic compartment due to the free energy gradient. Lastly, the holes and the electrons are consumed to promote oxidation (H_2_O, HX, etc.) and reduction (H^+^, CO_2_, etc.) pathways. One of the greatest advantages of the DSPECs is the employment of a sensitizer able to collect the visible photons precluded to the wide band-gap n-type semiconductors, such as TiO_2_ [29], SnO_2_ [30], and ZnO [31] (some examples will be described in Paragraph 4). Considering that the typical semiconductor nanoparticles dimensions are on the order of 20–30 nm, with an overall thickness ranging between 3–8 µm and sensitized with a monolayer of dye (ε ≈ 10^4^–10^5^ M^−1^ cm^−1^), this translates into a light-harvesting capacity higher than 90% on the maximum absorption [32,33].

An opposite situation (Figure 3b) can be realized in the presence of a photocathode in which the electrode is decorated with a sensitized p-type based-semiconductor like NiO or, more recently, the promising Cu(In,Ga)Se_2_ (CIGS) [34]. After the absorption of sunlight with the consequent generation of the dye’s excited state, the holes will be injected into the valence band (VB) of the mesoporous substrate and the reduced sensitizer is restored to its ground state by transferring an electron to promote the hydrogen evolution. The photoinjected holes, instead, migrate through the back contact to the anodic counterpart to promote the oxidation reaction.

The quasi-Fermi level is an important parameter that determines the thermodynamics of the cell and describes the capability to lead to the desired fuels. For instance, in order to promote hydrogen evolution without applying an external bias, the quasi-Fermi level of the photoanode, which is in equilibrium with the Fermi level of the cathode, should be more negative than the formal potential for the hydrogen evolution ($E_{H^{+}/H_{2}}^{{}^{\circ}}=0\;V$ vs. RHE). If this requirement is not met, an external bias is needed to enable the desired reaction (Figure 4). In particular, this voltage is applied by an external source, and it should also provide the cathodic overvoltage necessary to sustain the current flow. In a first approximation, this external bias does not affect the redox potential of the sensitized surface and thus, the photoproduced holes are able to prompt for the anodic reaction, such as water oxidation ($E_{O_{2}/H_{2}O}^{{}^{\circ}}=1.23\;V$ vs. RHE).

In order to avoid the external bias, a DSPEC should be realized by coupling an n-type semiconductor to a p-type substrate, realizing an n-p junction (Figure 5), in which the two compartments are able to promote reactions in consideration of their own photovoltage. This setup can be represented as a photochemical diode [35]. In all the above-described situations, an effective approach to collect the products is to physically separate the anodic and cathodic compartments with an ion-permeable membrane (PEM), the most common of these is the Nafion membrane, which is able to exchange protons from the anode to the cathode. This separation is required to avoid the reduction in the oxidized products at the cathode and vice versa (Figure 5).

## 3. Efficiency Parameters

The most important parameter that describes the efficiency of a DSPEC exposed to a broadband solar irradiance, such as the Air Mass 1.5 Global illumination (AM 1.5 G), is the solar-to-hydrogen conversion efficiency (STH). This parameter should be employed in zero-bias conditions, meaning when the applied voltage between the working and the counter electrode is zero, and the process is sustained solely by sunlight. Both the working and the counter electrode should be maintained in the same pH conditions to avoid a Nernstian gradient and the electrolyte must not contain any sacrificial agent that could affect the water-splitting performances [36]. STH can be defined as follows:(1)$$STH=\frac{\left|J_{sc}\left(mA{\;\mathrm{cm}}^{-2}\right)\right|\times \left(1.23\;V\right)\times \eta_{F}}{P_{total}\left(mW{\;\mathrm{cm}}^{-2}\right)}$$
where J_sc_ is the short circuit photocurrent density, 1.23 V represents the thermodynamic water splitting potential at 25 °C and pH 0, η_F_ is the faradic efficiency related to the hydrogen production, and P_total_ represents the incident illumination power intensity in mW cm^−2^. The STH is no longer valid when a bias between the working and the counter electrode is applied because the energy loss due to the need or the required overvoltage should also be taken into account. The applied bias photon-to-current efficiency is defined as follows:(2)$$ABPE=\frac{\left|J_{ph}\left(mA{\;\mathrm{cm}}^{-2}\right)\right|\times \left(1.23-\left|V_{b}\right|\right)\left(V\right)x\eta_{F}}{P_{total}\left(mW{\;\mathrm{cm}}^{-2}\right)}$$
where J_ph_ represents the photocurrent density obtained under an applied bias V_b_. Certain precautions are still applicable, such as avoiding the use of a sacrificial donor as well as maintaining the same pH in the two compartments of the cell. Another important figure of merit of the DSPECs is the incident photon-to-current conversion efficiency (IPCE%) that describes the photocurrent recorded per incident photon flux as a function of the monochromatic illumination (action spectrum) and can be described as follows:(3)$$IPCE\left(\lambda \right)=\frac{n_{e}}{n_{h\nu }}=\frac{hc}{e}\frac{J}{\lambda P\left(\lambda \right)}$$
where h is the Planck’s constant, c is the speed of light in vacuo (m s^−1^), *e* is the electron charge (C), J is the photocurrent density (A m^−2^), λ is the photon wavelength (m), and P(λ) is the radiant power density (W m^−2^). Generally, it is important to express the IPCE as a function of the key steps of the device working principle as follows:(4)$$IPCE\left(\lambda \right)=LHE\left(\lambda \right)\phi_{inj}\eta_{c}$$
where LHE is the light harvesting efficiency, Φ_inj_ is the injection quantum yield of the photogenerated electrons into the CB, and η_c_ represents the electron collection efficiency at the photoanode. This former parameter can be expressed as follows:LHE = 1 − 10^−A(λ)^
(5)
where the absorbance A can be made explicit as 1000 ε Γ with ε the molar extinction coefficient (M^−1^ cm^−1^) and Γ the dye surface concentration (mol cm^−2^). The photocurrent density (J_ph_) generated by a DSPEC device can be extrapolated by integrating the IPCE over the entire set of wavelengths as in Equation (6):(6)$$J_{ph}=\frac{e}{hc}\overset{+\infty }{\underset{0}{\int }}IPCE\left(\lambda \right)\;\lambda \;P\left(\lambda \right)d\lambda$$

The photocurrent density extrapolable from a DSPEC device can be also derived from the diode equation [37] as follows:(7)$$J=J_{ph}-J_{0}\left(exp\left(-\frac{eV_{j}}{mkT}\right)-1\right)$$
where J_ph_ is the limiting current density, V_j_ = V + iR represents the iR drop of the applied potential across the cell, and m is the ideality factor of the diode characterized by the J_0_ dark current. In particular, it should be highlighted that J_0_, m, and R depend on the specific chemical nature of the system, and they also describe the quality of the cell components, such as the mesoporous semiconductor characteristics, the ohmic contacts, and the conductivity of the solution.

## 4. Sensitizers and Surface Binding and Stabilization Strategies

To collect the visible light precluded by the wide band-gap semiconductors, the sensitizers represent one of the most important components of the DSPECs. To ensure the optimal functioning of the device, thereby translating into higher efficiencies, an ideal chromophore should exhibit some strict requirements. For instance, a good red response is enormously welcome since it can lead to efficient light harvesting by collecting the low-energy photons of the visible spectrum [33]. Large values for the driving force for the charge injection (ΔG_inj_ = |E_ox_^*^ − E_fb_|, where E_fb_ represents the flat band potential of the semiconductor) are essential to maximizing the injection quantum yield (Φ_inj_). Thus, it is imperative that the excited state oxidation potential (E_ox_^*^) of the sensitizer is more negative with respect to the metal oxide’s Fermi level, as depicted in the scheme of Figure 6 [32]. Efficient hole transfer to the target substrate is another important aspect and, from this perspective, the ground state oxidation potential of the sensitizer (E_ox_^(s/s+)^) must be larger than the oxidation potential of the reduced target molecule (E_ox_^(substrate)^) (e.g., water, HX, organic substrates, etc.), being ΔE = E_ox_^(s/s+)^ − E_ox_^(substrate)^ > 0 [38]. Additionally, stable anchoring groups are necessary to avoid hydrolytic detachment of the dye from the surface and, as expected, the sensitizers must be chemically and electrochemically stable to face several working cycles [39].

Since the pioneering contribution of Mallouk, ruthenium-polypyridine complexes [19] exhibit the best characteristics as light harvesters owing to their rather broad absorption extending between the blue-green portion of the visible spectrum and their relatively high molar extinction coefficient (ε ≈ 10^4^ M^−1^ cm^−1^). Ruthenium complexes are also characterized by a lifetime on the order of hundreds of ns, by electrochemical stability, and by ground and excited state oxidation potential, which matches the required thermodynamics (see Figure 6). Spectroscopic and electrochemical properties can be fine-tuned by introducing, for instance, different electron donating or electron-withdrawing functional groups on the ancillary ligands, typically bipyridines or terpyridines [40,41]. Electron injection is thermodynamically feasible from both the lowest singlet (^1^MLCT) and triplet (^3^MLCT) states but, in order to maximize the charge injection rate, the electronic coupling between the sensitizer and the substrate should be established through the formation of covalent bonds. Due to its relative stability and easy synthesis, the typical anchoring group adopted for classic DSSC devices consists of carboxylic moieties directly linked to the aromatic system of the ligand [32]. However, the resulting ester covalent bond between the anchoring moieties and the metal oxide can be easily cleaved in alkaline or even neutral conditions, leading to the desorption of the dye and the consequent decay of performance [42,43,44,45,46]. Among several alternative strategies that have been explored, studies that have focused on comparing the carboxylic (–COOH) and phosphonic (–PO_3_H_2_) anchoring groups are worth noting. For instance, a contribution realized by the Group of T. Meyer [47] highlighted a 90% of desorption in a water medium at pH 5.7 for complexes bearing -COOH against only 30% when –PO_3_H_2_ are employed. The group of Choi [42,43] investigated the effect of the number and the kind of grafting units in two families of ruthenium complexes bearing carboxylic (entries **1**, **2**, and **3** in Table 1) and phosphonic (**4**, **5,** and **6**) moieties, respectively. In particular, –PO_3_H_2_ bearing dyes resulted in a stable absorption over higher pH and the number of anchoring groups resulted in being less crucial than in the case of the -COOH-based analogs, owing to the strong bond exhibited by phosphonic groups with the TiO_2_. Despite the fact that the -PO_3_H_2_ anchoring group seems to be a better choice, the long-term stability of Ru-based complexes was achieved by employing dyed-surface passivation strategies, such as hydrophobic polymer overlayers [48], electro-assembly [49,50,51,52], siloxane [53,54] or silatrane surface bindings [55,56,57] or, more in general, hydrophobic environments [58]. Atomic layer deposition (ADL) to realize an aluminum oxide (Al_2_O_3_) or a titanium dioxide (TiO_2_) overlayer, starting from Al(CH_3_)_3_ or TiCl_4_ precursors, respectively, was found to represent an efficient way to prevent the hydrolysis of the ester binding mode. T. Meyer analyzed the stability of a water oxidation catalyst [Ru(Mebimpy)(4,4′-(PO_3_H_2_CH_2_)_2_bpy)-(OH_2_)]^2+^ (Mebimpy = 2,6-bis(1-methyl-1H-benzo[d]-imidazol-2-yl)pyridine) (**7**) with and without ALD overlayer on an ITO electrode, obtaining a stable film over 2.8 h in a pH 11 water medium [59]. The same group employed a “mummy” strategy to stabilize chromophore–catalyst structures (Figure 7a), obtaining light-assisted water oxidation stable over 6 h photolysis (at pH 8) [60]. This approach consisted in loading **4** onto *nano*TiO_2_ or *nano*ITO followed by an ALD overlayer of a 1.5 nm thick Al(CH_3_)_3_/H_2_O film. The catalyst **7** was then loaded onto the ALD overlayer and, subsequently, a final coating was realized in a similar fashion by increasing the total thickness to ~3 nm by “mummifying” the chromophore–catalyst assembly. 

An interesting strategy to achieve a stable linkage in a basic medium consists of employing pyridyl (py) anchoring moieties [61] that are known to form coordination bonds with surface Ti^4+^ ions, unlike the -COOH and -PO_3_H_2_ which interact with the Bronsted acid sites (i.e., the exposed OH groups) [62,63,64]. Indeed, stability tests in the electrolysis medium at pH 5 yielded negligible desorption of **8** (pyridyl linkage) at least for 2 h whereas, in the same conditions, **9** (-COOH linkage) and **10** (PO_3_H_2_ linkage) (Figure 7b) resulted in almost quantitative or 30% detachment, respectively, after only ≈10 min. In addition, **8** was stable in a neutral (pH 7) or basic (pH 9) aqueous buffer solution, clearly suggesting that the pyridyl anchoring moiety is likely a more suitable choice for photocatalytic applications under a water medium. A recent strategy was achieved by Mallouk by employing a phosphonate oligomeric ruthenium dye (**11**) [65], which resulted in a dramatically improved electrode adhesion in the range of pH between 4 and 7.8. The authors highlighted a faster cross-surface hole diffusion for the multinuclear species (≈ one order of magnitude) if compared to the monomeric one. It is important to note that this parameter was identified as one of the limiting factors to reach a higher turnover number [66,67,68]. 

Another approach to reach a stable system in a basic medium was proposed by the group of G. Meyer, who compared the diazonium group (**12**) with the classic -COOH (**13**) and -PO_3_H_2_ (**14**) sites as grafting agents for ruthenium complexes, as depicted in Figure 8 [69]. As for **12**, it was stable over 6 h in saturated NaOH while **14** desorbed by more than 50% in 1 min at pH 7 and **13** detached immediately at pH > 5. In addition, **12** exhibited a stable photocurrent response at pH 12 for 4 h confirming the stability of this new anchoring group in a basic medium. The same experiment was extended in acid conditions where **14** is highly stable: the photocurrent response was significantly higher for the phosphonic dye with respect to **12**; this is ascribable to a less efficient electronic injection of the latter, which arises from the weaker electronic coupling with the Ti sites and/or a fast-back electron transfer.

Other transition elements were explored as metal centers in coordination compounds in the framework of n-type sensitization such as Ni(II) [70,71], Cu(I) [72], Zn(II) [71], or, more recently, Fe(II) [73,74].

Porphyrins represent another promising sensitizers family [75,76] owing to their high molar extinction coefficient (>10^5^ M^−1^ cm^−1^ for the Soret band) over the high and low energy portion of the visible spectrum (i.e., Soret and Q bands) and owing to their redox properties well suited for DSPECs applications. In fact, the singlet ground state (S_0_) is generally higher in energy than the corresponding state of Ru complexes, resulting in stronger oxidizing capabilities by the photo-oxidized sensitizer (S^+^). The electron injection into the semiconductor’s conduction band is favorable from the Singlet excited states (S_1_ and S_2_) because the Triplet state (T_1_) is closer in energy or even lower than that of the CB (Figure 9). As in the case of the Ru-based sensitizers, the electrochemical properties can be tuned by varying the peripheral substituents [77]. 

Regarding the covalent attachment of porphyrin sensitizers, -COOH [78], -PO_3_H_2_ [79], -CNCOOH [75], and -py [80] have been explored to reach high stability, excited state directionality, and good electronic coupling. In particular, T. Meyer investigated the absorbance of electrodes sensitized with -COOH- and -PO_3_H_2_-based porphyrins every 15 min for 16 h in 0.1 M HClO_4_ [79]. A desorption of 80% and just 10% for -COOH or -PO_3_H_2_ grafting moieties, respectively, was observed, further confirming the improved adhesion provided by the phosphonate groups with respect to carboxylic functions. In the framework of HBr oxidation, Orbelli et al. [75] synthesized three perfluorinated Zn porphyrins (**15**, **16**, and **17**) bearing pentafluorophenyl electron-withdrawing groups to increase the ground-state oxidation potential. Moreover, sensitizers differed for the nature and position of the π–conjugate linker between the core and anchoring group, tasked to bind the metal oxide characterized by both carboxylic and cyanoacrylic groups. The latter resulted in the best performer among the series owing to the strongest electron-withdrawing properties that led to the best coupling and directionality for the electron injection. Since they worked in acidic medium, the photocatalytic performances were not affected by the stability of both the employed anchoring groups, which is not critical in these conditions. Among the macrocycles-based chromophores, phthalocyanines (Pcs) are robust and intensely colored (blue pigments) with high chemical, thermal, and light stability. These interesting features suggested this class of compounds as promising light harvester in the framework of solar cells [81,82,83].

However, despite the high performances of the described classes of compounds, ruthenium polypyridyl complexes are characterized by the main disadvantages of being rare, expensive, and toxic and by having a molar extinction coefficient that steeply decreases for λ > 500 nm. On the other hand, a relevant drawback when it comes to porphyrin dyes is related to the fact that their synthesis involves a multi-step route with a relatively low overall yield [84,85]. These limitations can be sorted out by employing metal-free organic dyes owing to their low cost, easiness of tuning their molecular orbital energetics by employing electron donating or withdrawing moieties, and also considering their extremely high molar extinction coefficient. The most widespread structure employs a donor–π bridge–acceptor (D-π-A) design which represents a class of compounds successfully analyzed in the framework of the regenerative cells [86]. After a few contributions related to the employment of perylene diimide (PDI), F. Li et al. reported one of the first applications of an organic-based sensitizer tandem cell. In detail, 4-(Diphenylamino)phenylcyanoacrylic acid (**18**), a D-π-A dye, was applied for the photoanodic compartment coupled with the well-known (4-(Bis-(4-[5-(2,2-dicyano-vinyl)-thiophene-2-yl]-phenyl)-amino)-benzoic acid) (**19**) for the photocathode side in which the photoanode was able to produce 300 µA cm^−2^ at pH 7 [87]. It is important to highlight the attempts to prompt the water splitting by employing 4,4-difluoro-4-bora-3a,4a-diaza-s-indacene (BODIPY) dyes given their high molar extinction coefficient and photostability, which translates into promising performances [88,89]. After having adopted a π-extended BODIPY with a near infrared response for DSSC applications [90,91] and for bulk heterojunction solar cells [92], Kubo et al. synthesized **20**, which is characterized by a λ_max_ at around 687 nm (λ_onset_ ≈ 728 nm) with an ε of 1.09 × 10^5^ M^−1^ cm^−1^. This molecular system was used for water-splitting applications [93]. Thanks to the spectroscopic, electrochemical, and photoelectrochemical analysis, they were able to demonstrate the capability of **20** to collect the NIR region and the ability to provide the water oxidation by applying a low overpotential. A similar approach was followed by T. Meyer that sensitized a SnO_2_/TiO_2_ photoanode with a series of BODIPY for the oxidation of hydroquinone and benzyl alcohol. The best performing, namely **21** (*N*,*N*′-Difluoroboryl-1,7-dimethyl-3,5-bis(4-(3,6-di-tert-butylcarbazol-9-ylstyryl)phenyl)-8-[4-(dihydroxyphosphoryl)phenyl]dipyrrin), was able to reach a stable photocurrent of 200 µA cm^−2^ in the case of the oxidation of the former substrate and a current of 35 µA cm^−2^ in the case of the latter (Figure 10) [94].

**Table 1 molecules-29-00293-t001:** Structures of the molecules reported in the main text. S and C refer to a sensitizer or a catalyst, respectively.

Structures												
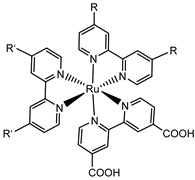	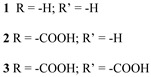			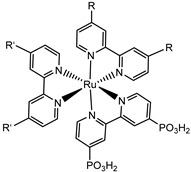		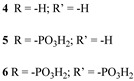		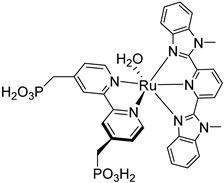				7
S [42,43,95]				S [42,43,60,96,97,98,99]				C [59,60,97]				
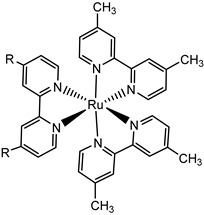	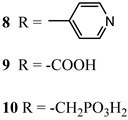			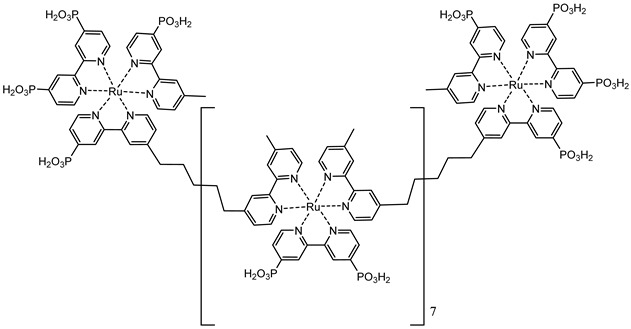								11
S [61]				S [65]								
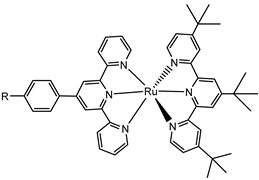		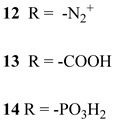		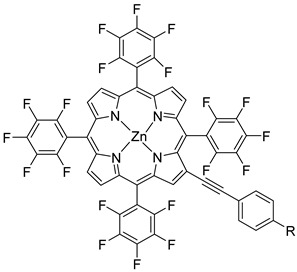					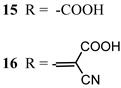			
S [69]				S [75]								
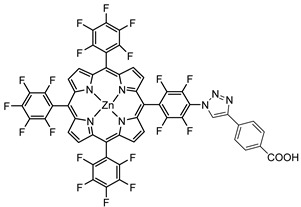			17	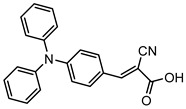			18	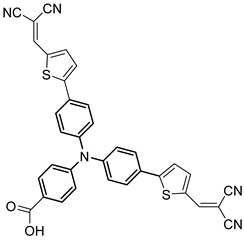			19	
S [75]				S [87]				S [87]				
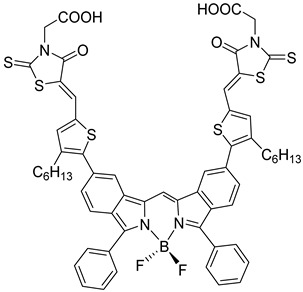				20	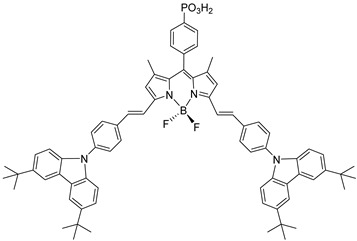					21		
S [93]					S [94]							

## 5. Water Splitting

The overall water splitting process can be described by the following reactions:(8)$$\mathrm{ANODE}:\;\;\;H_{2}O\to \frac{1}{2}O_{2}+2H^{+}+2e^{-}$$



(9)
$$\mathrm{CATHODE}:\;\;\;2H^{+}+2e^{-}\to {\;H}_{2}$$



(10)$$\mathrm{TOTAL}:\;\;\;H_{2}O\to \;\frac{1}{2}O_{2}+{\;H}_{2}$$
where Equations (8) and (9) are the oxygen-evolving reaction (OER) and the hydrogen-evolving reaction (HER), respectively. The kinetic bottleneck of water splitting arises from the requirement of four-hole equivalents to produce one mole of O_2_ [100]. The net reaction described in Equation (10) can store 237 kJ per mole of H_2_, which corresponds to an electromotive force for the cell reaction (E_rxn_) of −1.23 V based on $\Delta G=-nFE_{rxn}$ where G is the free Gibbs energy, F is the Faraday constant, and *n* is the number of involved electrons, (*n* = 2 for Equation (10)) [100,101]. The energy required to enable the migration of an electron from the anodic to the cathodic compartment of the DSPEC is on the order of 1.23 eV and can be provided by electromagnetic radiation with a wavelength smaller than 1010 nm according to the Plank–Einstein relation E = hν [102]. However, due to overpotential requirements for promoting H_2_ and O_2_ production, along with thermodynamic limitations, the ∆G increases from 1.23 eV to values higher than 2.03 eV (∆G + U_loss_). Here, U_loss_ corresponds to the energy lost per photon and can be considered as equal to at least 800 meV [103]. Consequently, the limiting absorption edge can be generally set to λ < 610 nm [103,104].

### 5.1. Water Oxidation Catalyst (WOC)

The complex kinetic route that drives the water oxidation involves multi-electron steps and would result in exceedingly high activation barriers for the direct hole transfer from the photo-oxidized chromophore to water molecules, thereby resulting in predominant back electron recombination. Thus, the water oxidation catalyst (WOC) plays an important role in boosting the performances of DSPECs. Molecular complexes offer advantages such as a fast rate for electron transfer (ET), redox and synthetic tunability, a relatively fast rate for water oxidation, and improved catalytic efficiency, especially when scarce metals are used. Among the catalysts, ruthenium complexes are widely studied owing to their well-known synthetic route, strong metal-to-ligand bonds, rapid rates of oxygen evolution ensuring system stability, and well-defined oxidation states (Ru^II^, Ru^III^, Ru^IV^, and Ru^V^) able to facilitate the water oxidation. Some representative Ru-based WOC are depicted in Table 2.

In 1982, T. Meyer and co-workers introduced the first homogeneous WOC knows as blue dimer, which is a ruthenium polypyridyl dinuclear complex characterized by a Ru-O-Ru bridge and the coordination of two H_2_O molecules to each Ru center (Table 2-I) [105,106]. The μ-oxo bridge plays an important role in stabilizing the complexes in their high oxidation states through electronic delocalization, facilitated by the strong coupling between the two metal centers. The proposed mechanism involves four proton coupled electron transfer (PCET) steps that give access to a high-valent Ru(V,V) intermediate (*intermediate 2*, Figure 11), which is subsequently attacked by H_2_O. This yields the formation of a hydroperoxo species (*intermediate 4*, Figure 11) that is intramolecularly oxidized by the second metal core, ultimately resulting in the release of O_2_ (*intermediate 6*, Figure 11) [107].

The instability of this class of WOCs is mainly due to the cleavage of the μ-oxo bridge, forming the single non-active monomeric system, which generally led to relatively low performances towards water oxidation. To sort out this problem, Liobet et al. [108] reported the first example of a dinuclear ruthenium complex (Ru-Hbpp = [Ru_2_(OH_2_)_2_(bpp)(tpy)_2_]^2+^ where Hbpp = 2,2′-(1H-pyrazole-3,5-diyl)dipyridine) capable of driving water oxidation in the absence of the Ru-O-Ru bridge (Table 2-II). The oxygen evolution rate of this complex was found to be three times larger than that of the blue dimer. Mimicking the multimetallic core of the natural OEC, all the first examples of WOC were based on multiple metal cores, spreading the idea that at least two metal centers were needed to prompt the water oxidation. In 2005, Thummel’s group reported the first evidence of a single-site catalyst capable of driving the OER (Table 2-III) [109]. This study was then corroborated by Meyer and co-workers in 2008 who proposed a well-defined mechanism involving the Ru(V) state, paving the way for a series of stable and efficient single site catalysts (Table 2-IV) [110,111]. One of the main drawbacks of the employment of the aforementioned neutral and π-accepting ligands-based WOC was found in the high oxidation potential necessary to reach the highly oxidized Ru(IV) and Ru(V) states, which are generally poorly aligned with the chromophore ground state level. In fact, Ce(IV) is often required as a triggering agent for oxygen production. Inspired by studies mimicking the OEC in PSII, which showed that phenolate and carboxylate ligands can efficiently stabilize high-valent Mn states, Sun and co-workers were the first to introduce carboxylate moieties to access high-valent Ru=O under lower potential. Their first example reported on a trans-dimeric complex coordinated by a 3,6-bis(6-carboxypyrid-2-yl)pyridazine (H_2_cppd) ligand, which is a modified pyridazine-based polypyridyl equatorial backbone containing two carboxylate groups (Table 2-V) [112]. This was followed by the cis-conformation (Table 2-VI) [113] and by a Ru(bda)(pic)_2_ monomeric system (where bda = 2,2′-bipyridine-6,6′-dicarboxylic acid) bearing two carboxylate groups in the equatorial ligands (Table 2-VII) [114]. Thanks to the strong electron-donating capabilities of the carboxylate moieties, the Ru(bda)(pic)_2_ exhibited a significantly lower oxidation potential compared to Ru complexes prepared with neutral ligands. The promising performances, coupled with the easy tunability of the axial ligands of this latter class of compounds, attracted an extensive attention and efforts from the scientific community, leading the application of the Ru-bda WOCs for efficient DSPECs [107,115,116].

### 5.2. Examples of Water splitting

As pioneered by Mallouk and co-workers [19], Ru(II) polypyridine complexes emerged as a very promising class of sensitizers for photoelectrochemical cells. Mallouk employed a Ru(II) tris(bipyridine) heteroleptic derivative (**22** inTable 3) to oxidize water (Figure 12) in conjunction with an IrO_2_·nH_2_O WOC, which was synthesized and characterized in detail by Harriman et al. many years before [117]. Prior to supporting the sensitizer–catalyst system onto TiO_2_, it was analyzed in contact with a 1 M Na_2_S_2_O_8_ solution, acting as an electron acceptor. Upon laser excitation at 532 nm, an electron transfer from the excited state of the dye (**22***) to the acceptor S_2_O_8_^2−^ takes place. The regeneration of the reduced form of **22** is thus driven by an electron transfer from the nanoparticles of IrO_2_·nH_2_O WOC, with a time constant of about 2 ms. When the system was supported onto TiO_2_, the oxidative quenching of **22** occurred by injecting electrons into the CB of the semiconductor. However, it was observed that the back-electron transfer was approximately one order of magnitude faster compared to the regeneration process facilitated by IrO_2_·nH_2_O, thereby limiting the performances of the device.

The photocurrent density–voltage profile in a buffered solution (pH 5.75) containing 30 mM Na_2_SiF_6_ and 500 mM Na_2_SO_4_ revealed an open circuit potential of −325 mV vs. Ag/AgCl along with a stable photocurrent of 30 μA. In the presence of an undyed TiO_2_, only a marginal current of ≈1–2 μA was observed. The authors were able to successfully describe the water-splitting reaction in the presence of visible light. However, they recorded a photoanodic oxygen generation of roughly 20%, which was mainly attributed to the slow electron transfer from the WOC to **22***. Based on the Mn_4_Ca active site into the PSII, Rüttinger et al. synthesized a [Mn_4_O_4_L_6_]^+^ (or “cubium”, L = (MeOPh)_2_PO^2−^) (**23**) WOC [118,119]. To ensure water solubility, the WOC was then supported onto a thin Nafion membrane [120]. Subsequently, the WOC has also been investigated in DSPEC in combination with the sensitizer **1** (Figure 13a), which is characterized by a 2–300 meV driving force for the oxidation of the WOC (**23**) (the system is reported as: **23**-Nafion/**1**-TiO_2_) [95].

The undyed TiO_2_ generated a photocurrent lower than 1 µA cm^−2^ (Figure 13b, black line), a slightly higher photocurrent of ≈5 µA cm^−2^ (Figure 13b, red line) for the Nafion/**1**-TiO_2_ system, a comparable value for **23**-Nafion/TiO_2_ and a significantly higher and sustained value for the complete **23**-Nafion/**1**-TiO_2_ of ≈15 µA cm^−2^. An IPCE of 1.7% was observed (Figure 13b bottom) and it was consistent with the shape of the MLCT band of **1**, confirming its capabilities as absorbing and injecting species. A similar approach was then realized by Li et al., leading to a significantly higher photocurrent density [96], by employing the sensitizer **4** and a **24**-based WOC [114]. 

Since the catalytic oxidation driven by **24** is pH dependent (the onset for water oxidation was found to be 1.5 V at pH 1 and 0.98 V at pH 7); to prevent the catalytic process of **24** from shifting to a higher potential than the E_1/2_ of **4**, the sulfonic acid groups in the Nafion membrane were neutralized before immobilizing the complex. The cationic WOC was generated with Ce^IV^ before intercalating the species into the membrane. The transient photocurrent densities are reported in Figure 14a. In 0.1 M Na_2_SO_4_ medium, the TiO_2_-Nafion-complex **24** system generated a negligible photocurrent (blue curve). When the WOC was replaced by the sensitizer, an initial photocurrent was detected which rapidly decayed. However, the entire TiO_2_-**4**-Nafion-complex **24** system led to a significant enhancement in the current, which was ascribed to the electron transfer from the catalyst to the photogenerated oxidized sensitizer. However, the classical overshootings ascribable to electron recombination were highlighted. The authors also analyzed the performances of the DSPEC under different pH conditions of the membrane, from 7 to 9.8, as reported in Figure 14b. In the latter condition, the current decay was lower and this was attributed to the rapid proton release occurring during the OER, thereby affecting the catalytic capabilities of **24** [121]. The current decay could also be ascribed to possible charge recombination of the photoinjected electrons onto the CB with the oxidized **24 [95]**. The catalytic properties of **24** were analyzed by examining the produced oxygen: in the absence of the WOC, no O_2_ was detected, while ca. 9-fold higher quantity was found for the complete system compared to that characterized by the absence of the sensitizer (140 nmoml^−1^:16 nmol ml^−1^ TiO_2_-**4**-Nafion-complex **24**:TiO_2_-Nafion-complex **24**), resulting in a turnover number (TON) of 16 and a turnover frequency (TOF) of 27/h. 

Based on earlier evidence of Mallouk regarding the capabilities of the Ru(II) polypyridyl-IrO_x_ system in performing the water splitting, Michaux et al. described the employment of **5** in a FTO/*nano*ITO/TiO_2_-**5**-IrO_x_ system where IrO_x_ acted as a nanoparticles (NPs)-based WOC for the OER [98]. The performances of the entire system and of the components themselves on a mesoporous TiO_2_ electrode have been analyzed by means of the current–time profile. The approximate three-fold enhancement of the photocurrent when moving from the single components to the semiconductor-dye-WOC assembly pointed out the efficient synergetic catalytic effect in promoting the water splitting. At the time of the contribution, the current was three times larger than the value previously reported in the literature owing to the uncapped nature of the IrO_x_ NPs, which were shown to enhance the water oxidation catalysis [69,122,123]. When the dye-NP system was supported onto *nano*ITO/TiO_2_ core shell electrode, the photocurrent was significantly improved, primarily due to the decrease in back electron transfer of the core/shell material. Figure 15a reports the extracted photocurrent density of two families of the TiO_2_ shell, with different thicknesses, as a function of the applied bias. In particular, the two different TiO_2_ thicknesses were obtained by varying the number of ALD TiO_2_ cycles, from 50 (red dots) to 100 (blue dots). The 6.6 nm thick TiO_2_ shell (100 cycles—blue dots) exhibited the best performance. Specifically, the higher the applied bias, the more consistent the extracted photocurrent, suggesting a fast electron transfer along the system.

The 2 h photolysis experiments (Figure 15b) highlighted a significant loss in the performance over time, likely due to desorption of the -PO_3_H_2_-based sensitizer because of the hydrolysis of the ester bond in the pH conditions of the NaSiF_6_ buffer (pH 5.8). This problem was sorted out, or at least minimized, by employing an ALD overlayer carried out by depositing 10 TiO_2_ cycles, with a resulted thickness of less than 1 nm, before incorporating the iridium-based NPs onto the photoanodes. While the non-stabilized photoanode exhibited a photocurrent lower than the dark current (−97 µA cm^−2^), the ALD-based electrode demonstrated a sustained current of 110 µA cm^−2^, indicating a significant improvement in the stability. Promising results were achieved by Takijiri et al. with the aforementioned strategy (i.e., pyridyl (py) anchoring moieties) [61], employing the sensitizer **8** to facilitate water reduction in the presence of EDTA as sacrificial agent. The chopped photocurrent density–voltage profile in a standard three-electrode cell, filled with 0.1 M acetate buffer/30 mM EDTA/0.1 NaClO_4_ aqueous solution, revealed an anodic photocurrent of 1.1 mA cm^−2^. Long-term electrolysis demonstrated a stable photocurrent with only a 7% loss over 100 min, accompanied by nearly quantitative Faradaic efficiency (FE (%)) for the H_2_ production. Recently, based on the improved stability highlighted by the -py-based anchoring moieties, T. Meyer has reported an extensive study concerning the role of the grafting moieties of a series of derivatized Ru(bda)-based WOC (bda = 2,2-bipyridine-6,6-dicarboxylic acid) in conjunction with the sensitizer **4** (Table 1), employed to drive the water oxidation with outstanding performances [99]. As depicted in Table 3, the bda-based WOC molecular systems **25**, **26** exhibit two different couple of ligands in their respective axial positions. In particular, two 4,4′-bipyridine units are used in **25** whereas the phosphonic acid-based ligands 4-pyridyl phosphonic acid was adopted to construct the complex **26**. 

The photoelectrochemical performances carried out in aqueous 0.5 M NaClO_4_ buffered at pH 5.8 revealed a 12-fold higher photocurrent density when the electrode was equipped with **25**, reaching 1.7 mA cm^−2^ (with an IPCE of 25%) compared to the 140 µA cm^−2^ exhibited by **26** (Figure 16a). To gain insight into the lower performances of **26** and to better clarify the role of the bridging moieties, the authors performed photoelectrochemical impedance spectroscopy measurements (Figure 16b). The typical Nyquist plot of a DSPEC is characterized by a semicircle arising from the recombination charge transfer resistance (R_rec_) at the interface between the TiO_2_|sensitizer and the WOC [124,125,126]. This parameter can be used to directly estimate the capability to counteract the back electron transfer, taking into account that the same amount of WOC is deposited onto the dyed electrode. Indeed, a slower recombination process was observed for **25** if compared to **26**, as confirmed by the larger recombination rate constant of 146 s^−1^ exhibited by the latter, unlike the former which yielded a value of 56 s^−1^ and highlighted the important role of the pyridyl anchoring moieties in inhibiting the back electron transfer. Based on these results, Meyer’s group introduced a third Ru-bda which bears two axial diethyl 3-(pyridin-4-yloxy)decyl-phosphonic acid ligands to construct the WOC **27** (see Table 3) with the aim to further separate the catalytic core from the oxidized surface and, thus, to reduce the back electron transfer. In fact, the recombination rate decreased to the smallest value among the series of 35 s^−1^. This translated into a photocurrent of 0.8 mA cm^−2^ for **27**, approximately a 6-fold difference from than that of **26**, but surprisingly this value was only half of the photocurrent exhibited by **25** (insert of Figure 16a). This result was ascribed to the formation of surface-bound μ-oxo-bridged dimers by **26** and **27** phosphonate-based catalysts, which exhibited a decreased reactivity towards water oxidation [127,128].

#### Molecular Sensitizer–Catalyst Assembly

The development of covalently linked sensitizer–WOC assemblies is one of the most important breakthroughs, primarily due to the favorable electron transfer kinetics arising from the closeness of the two components and the optimal stoichiometric ratio between them [129]. In 2015, T. Meyer reported a [(PO_3_H_2_)_2_bpy)_2_Ru(4-Mebpy-4′-bimpy)Ru(tpy)(OH_2_)]^4+^ chromophore–catalyst assembly (**28**) onto *nanoITO*–TiO_2_ (ALD deposited) core-shell, capable of driving the water oxidation (Figure 17a). To highlight both the ruthenium cores, the whole system can be shortened as FTO|*nano*ITO/TiO_2_-[Ru_a_^II^-Ru_b_^II^-OH_2_]^4+^ [130]. The MLCT transition of the sensitizer unit in **28** generates the chromophore excited state followed by the charge injection into the CB of the semiconductor, resulting in the charge-separated state (*semiconductor(e^−^)-Dye(+)*). This triggers a series of electron transfer events between the two ruthenium cores. Transient absorption kinetics on the nanosecond timescale revealed an injection occurring within 20 ns after laser pulse; the bleach recovery, owing to the back electron transfer, was noticeably longer for **28** compared to the sensitizer alone, consistent with hole transfer from the chromophore to the WOC (Figure 17b). The artificial photosynthesis prompts three single-photon-single-electron excitation events for each oxidative cycle to reach the (Ru_a_^III^–Ru_b_^IV^=O)^5+^ active intermediate, which is able to undergo attacks by water. However, due to the back electron transfer on a μs to ms timescale and the requirement for three injection–oxidation cycles to activate the WOC, the active species is present only in trace amounts under ambient solar illumination conditions. Consequently, when **28** is loaded onto TiO_2_, the resulting photocurrent is negligible. To minimize the recombination losses, core-shell particles have been developed. Under monochromatic illumination, the FTO|*nanoITO*/TiO_2_-[Ru_a_^II^-Ru_b_^II^-OH_2_]^4+^ exhibited a photocurrent of ≈200 μA cm^−2^, which decreased to 20 μA cm^−2^ after 30 min illumination. In pH 4.6 acetate buffer medium, short-term experiments exhibited a stable current of 100 μA cm^−2^ under 91 mW cm^−2^ spectral irradiance with an applied bias of 0.2 V vs. NHE.

A slightly different approach was achieved by employing a SnO_2_/TiO_2_ core/shell structure, where the SnO_2_ core benefits from a CB more positive than TiO_2_ of ca. 400 meV, in conjunction with the molecular assembly **28** above described [131]. Once the electrons reach the core, the internal gradient at the SnO_2_/TiO_2_ interface prevents back electron transfer (Figure 18a).

In Figure 18b (insert), the photocurrent–density curves of the short-term experiments conducted on *nanoITO*/TiO_2_ and SnO_2_/TiO_2_ electrodes are depicted. Peaks of current on the order of 480 μA cm^−2^ were observed for SnO_2_/TiO_2_ (3.3 nm) core/shell photoanodes, which then dropped to 100 μA cm^−2^ after 10 sec electrolysis in 0.5 M LiClO_4_/20 mM acetate buffer at pH 4.6. With the employment of a 0.66 nm TiO_2_ overlayer, deposited by the ALD method, the maximum photocurrent density increased to 790 μA cm^−2^. The role of the overlayer was ascribed to the stabilization of the phosphonate groups on the surface without affecting the charge injection, which was facilitated by quantum mechanical tunneling from the excited state of the dye. It is noteworthy that the replacement of *nanoITO*/TiO_2_ with the SnO_2_/TiO_2_ system led to a five-fold increase in photocurrent. Long-term experiments revealed an impressive stabilization toward hydrolysis until pH 7 when the photoanode is equipped with the ALD overlayer (FTO|SnO_2_/TiO_2_(6.6 nm)|-[Ru_a_^II^-Ru_b_^II^-OH_2_]^4+^-(0.6 nm)TiO_2_) (Figure 18b). The gradual decrease in photocurrent was mainly ascribed to ligand loss by the Ru(III) form of the sensitizer in the assembly [45]. Generally, the realization of the sensitizer–WOC dyad suffers from multi-step synthesis, relatively low surface loading, and scarce versatility. Based on the contribution of Mallouk [132,133] and Haga [134,135] concerning the realization of multilayer alkyl diphosphonate films and multilayer Ru/polypyridyl assemblies, respectively, T. Meyer exploited a facile “layer-by-layer” approach by phosphonate/Zr^IV^ coordination linkages to realize sensitizer–WOC self-assembly, as displayed in Figure 19 [136]. 

The bi-layer molecular structure (**29**) (Figure 19) can be easily achieved in a stepwise manner by subsequently immersing the photoanode in solutions of the sensitizer, ZrOCl_2_, and either another sensitizer or the WOC. Transient absorption spectroscopy revealed the capability of the self-assembly system to undergo inside-to-outside electron transfer, facilitating hole migration from the sensitizer to the WOC for water splitting. The back electron transfer from the photoinjected electrons to the WOC^+^ was significantly slowed down in the case of the whole system, which is consistent with an efficient charge separation along the assembly. A similar approach was explored by Sun e co-workers, who realized a self-assembly dye-catalyst characterized by a Ru(bpy)_3_ sensitizer coordinated to a Ru-bda unit through a Zr(IV) bridge (**30**, see Table 3) [137]. Under 3 SUN irradiation and in pH 6.4 buffered aqueous media, this assembly was able to drive the OER with a stable photocurrent of 500 μA cm^−2^, corresponding to an IPCE of around 4%. Later, T. Meyer combined the effect of the SnO_2_/TiO_2_ core/shell structure with the efficient Ru-bda WOC (**31**, Table 3) linked to a series of Ru(II) polypyridyl chromophores (entries **5** and **6** in Table 1) via the Zr(IV) bridge [138]. The study compared the performances of these assemblies with that of co-loaded WOC on the surface. Under the best conditions, the performances of the two systems were essentially aligned. In the presence of the chromophore/assembly **6**-Zr-**31**, the plateau photocurrent of 1.45 mA cm^−2^ was found after 20 s photolysis, with an FE (%) of 74% for the O_2_ production, as determined by the collector–generator (C-G) technique. By combining all the knowledge described, a relatively simple mimetic of the components of natural PSII was recently reported (**32**) (Figure 20). It includes a mediator that reproduces the tyrosine-histidine redox couple, which is not commonly reported in water splitting-based systems [139]. This role was fulfilled by a triphenylamine (TPA) derivative acting as an electron transfer mediator between the oxidized dye and the WOC. In nature, tyrosine inhibits the back electron transfer and stabilizes the system by storing oxidative equivalents close to the WOC. In this contribution, the aim of the TPA was to act as a one-electron transfer mediator. TPA was covalently bonded to a phosphonate polypyridyl Ru(II) complex, which acted as the chromophore. The sensitizer was also linked to a Ru(II)-bda WOC, while the semiconductor was primarily covered by the SnO_2_/TiO_2_ core-shell. The system was analyzed with and without the TPA-based mediator.

Comparable photocurrent responses were found for both the systems; however, the chopped photocurrent density–time profiles (J-t) (Figure 21a) showed a higher initial current for the TPA-free electrode, which then decreased by over two factors. A similar behavior was found in the long-term experiments in pH 7.0 solution at 0.6 V vs. NHE (Figure 21b), in which in the initial stage the current was 40% higher than in the absence of the TPA moieties as a consequence of the faster hole transfer to the catalyst. However, after 3 h of electrolysis, the TPA-based systems yielded a photocurrent of 120 μA cm^−2^ while after 1 h, the control system without the mediator reported 85 μA cm^−2^. These data clearly highlight the importance of employing TPA-based moieties for system stability. By means of the C-G experiments, the authors reported an FE (%) of 83% and 75% for O_2_ production in the systems with and without the TPA mediator. All the photoelectrochemical details such as the type of sensitizer and catalyst, the photocurrent density, and the applied potential as well as the experimental conditions are summarized in Table 4.

**Table 3 molecules-29-00293-t003:** Structures of the molecules reported in the main text to drive water splitting. S represents a sensitizer and C a catalyst, while A represents an assembly.

Structures										
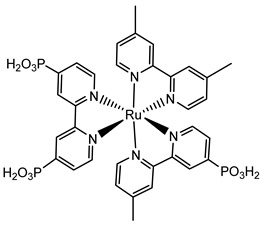			22	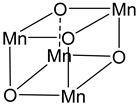		23	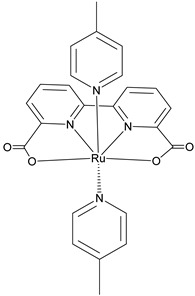		24	
S [19]				C [95,118,119,120]			C [96,114]			
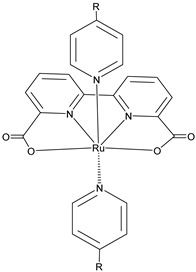	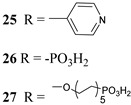			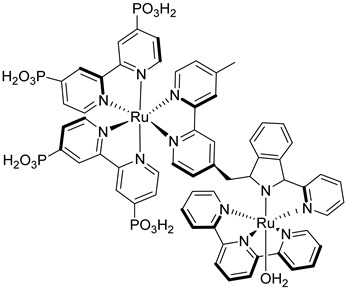				28		
C [99]				A [130]						
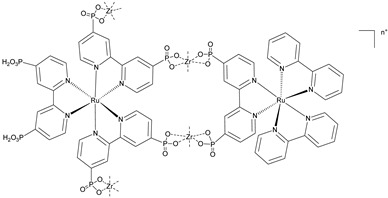				29	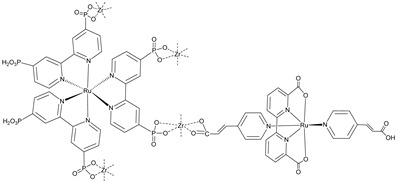					30
A [136]					A [137]					
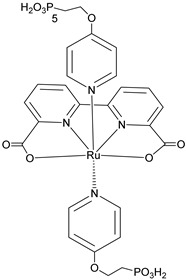		31		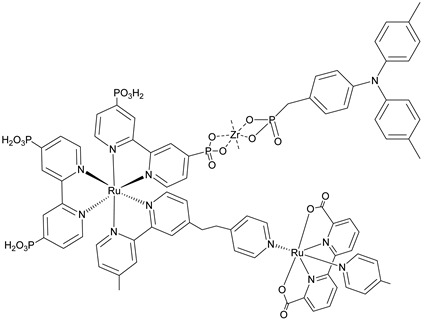						32
C [138]				A [139]						

## 6. Hydrobromic Acid Oxidation (HBr)

Since oxygen generated from water splitting results in a product with poor economic value, several attractive alternatives have been found. For instance, the hydrohalic acid (HX, X = Cl^−^ or Br^−^) splitting results in a promising inexpensive storage way for solar energy, which involves the reduction of H^+^ to H_2_ at the cathode and the concomitant oxidation of X^−^ to X_2_ at the anode, generating Cl_2_ or Br_2_ [23]. Among the HX family, hydrobromic acid revealed the largest theoretical solar-to-hydrogen efficiency. Once the solar energy is stored in Br_2_, it can be earned back using fuel cells [25,140,141,142] or redox flow batteries [27]. The electrochemistry of Br^-^ in aqueous medium has been extensively investigated [143] and the most relevant redox processes can be summarized by the following equations [143,144]:(11)$$3Br^{-}\to Br_{3}^{-}+2e^{-},\;E^{{}^{\circ}}=1.05\;V\;vs\;NHE$$
(12)$$2Br^{-}\to Br_{2}+2e^{-},\;E^{{}^{\circ}}=1.09\;V\;vs\;NHE$$
(13)$$Br_{2}+Br^{-}⇆Br_{3}^{-},K_{eq}=17\;\mathrm{at}\;298\;K$$
(14)$$Br_{2}+2e^{-}\to 2Br^{-},\;\;k_{rec}>10^{4}\;s^{-1}$$

From this series, the small equilibrium constant between Br_2_ and Br_3_^−^ in the presence of the Br^−^ anion, should be highlighted [145] as well as the fast recombination as a consequence of the back electron transfer kinetics of Br_2_ with the metal oxide surface [146]. The relatively high oxidation potential (1.05 V or 1.09 V) (Figure 22) of the Br^-^ species can also activate a [Ru(bda)(L_2_)] catalyst for water oxidation, resulting in the so-called bromine-assisted water splitting [144] where the most challenging step is the Ru^IV/V^ oxidation, with a potential close to 1 V for pH higher than 5.5 [147,148]. 

It should not be surprising that ruthenium-based complexes have also found interesting applications in bromide oxidation owing to their suitable redox and spectroscopic properties [149,150]. Concerning the heterogeneous catalysis, Li et al. reported one of the earliest examples of Br^−^ homogenous catalysis driven by Ru(II) complexes in acetone medium [149,150,151]. Later, Tsai et al. found an Ru(II) complex able to perform the Br^−^ oxidation in acetonitrile medium, generating the Ru(III) species by employing a sacrificial oxidant [152,153]. In 2017, T. and G. Meyer published a communication on heterogeneous catalysis, reporting the splitting of HBr driven by a [Ru(btfmb)_2_P]^2+^ species (btfmb = 4,4′-bis(trifluoromethyl)-2,2′-bipyridine and P = 2,2′-bipyridyl-4,4′-diphosphonic acid) (**33 Table 5**) loaded onto TiO_2_ (**33**|TiO_2_) or SnO_2_/TiO_2_ core-shell (**33**|CS)-based electrodes [154]. The photophysical properties of **33** were investigated by means of the nanosecond transient absorption measurements. A $\phi_{inj}$ of 57% was quantified by employing a comparative actionometry [155,156] using **4** as an actinometric standard, 50 ns after the laser excitation. Owing to the distribution of recombination rate constants motivated by the inherent inhomogeneity of the nanocrystalline mesoporous thin film, the 400 nm recombination dynamics was accurately modeled by the Kohlrausch–Williams–Watts (KWW) stretched exponential function. For **33**|CS systems, the recombination rate was found to be two orders of magnitude smaller if compared to the TiO_2_ counterpart **33**|TiO_2_. The introduction of Br^−^ into the electrolyte resulted in a regeneration efficiency ($\phi_{reg}$) of 99% and 41% for the CS and TiO_2_ systems, respectively, which is consistent with the slow recombination provided by the electron gradient arising in the core-shell structures.

The photocurrent density–voltage curves for **33** supported onto core-shell or TiO_2_ in contact with 1 M HBr are reported in Figure 23. In both cases, the onset of the photocurrent density was around 0.4 V and the core-shell system revealed an 8-fold higher photocurrent response compared to the TiO_2_ analog, reaching to a maximum photocurrent of 1.5 mA cm^−2^ for the former device. This is consistent with the expectation based on the transient absorption data concerning the lower charge recombination exhibited by the CS structure. This translated into an IPCE of 20% and an APCE of 24% at 460 nm under an applied potential of +0.6 V. The Faradaic efficiency measurement for the Br^-^ oxidation was carried out by means of the C-G (collector–generator) and absorption spectroscopy experiments. The latter was employed to determine the concentration of the Br_3_^−^ species through its high relativity molar extinction coefficient of 40,900 M^−1^ cm^−1^ at 266 nm. On the other hand, the C-G measurements were adopted to quantify the Br_2_ species, characterized by an ε of only 175 M^−1^ cm^−1^ at 390 nm. The C-G measurement revealed a Faradaic efficiency of 71 ± 5%, in perfect agreement with the value of 72%, which was spectroscopically recorded. Slama-Schwok et al. [157] indicated that Br^•^ radicals, generated in acidic water, may react with transient oxygen radicals formed from water that could potentially decrease the FE (%). 

In 2018, G. Meyer investigated the effect of varying the redox properties of the surface-bound photocatalyst from a new family of Ru(II) polypyridyl complexes towards HBr splitting. These complexes had a general formula [Ru^II^(LL)_2_P]^n+^ (where P is 2,2′-bipyridyl-4,4′-diphosphonic acid and LL is 2,2′-bipyridine (bpy), 4,4′-bis(trifluoromethyl)-2,2′-bipyridine (btfmb), 4,4′-bis[(trimethylamino)-methyl]-2,2′-bipyridine (tmam), and 2,2′-bipyrazine (bpz) (**4**, **33**, **34**, and **35**) [158]. 

To investigate the injection quantum yield, the charge recombination, and the regeneration efficiency, the photophysical analysis was carried out by means of the nanosecond transient absorption spectroscopy. The formation of the charge separated state (*Core/Shell(e^-^)/[Ru(III)(LL)_2_P]*) occurred within the laser pulse, resulting in a k_inj_ > 10^8^ s^−1^. $\Phi_{inj}\;$ was determined 30 ns after the 532 nm laser excitation, employing **4**|CS as an actionometric standard and sampling at the ground state/excited state isosbestic point. The values obtained were 1.00 for **4**|CS, 0.59 for **33**|CS, 0.77 for **34**|CS, and 0.09 for **35**|CS. Charge recombination was probed at the ground-state/excited-state isosbestic point near 400 nm for each Ru-based complexes loaded onto the core/shell. The average charge recombination rate constants (k_cr_) were determined by fitting the kinetics by means of the KWW function, resulting in values of 5.2 × 10^2^, 2.8 × 10^2^, and 8.9 × 10^2^ s^−1^ for **4**|CS, **33**|CS, and **34**|CS (Figure 24a), respectively. A bromide titration was performed to extrapolate the K_reg_ for the [Ru(III)(LL)_2_P] complexes in the presence of Br^-^; in particular, it was computed from the slope of the average rate constant (k_KWW_) versus bromide concentration for each compound (Figure 24a-insert). These data led to a regeneration efficiency $\Phi_{reg}$ spanning from 1.00 for **35**|CS to 0.037 to **4**|CS consistent with unfavorable energetics for bromide oxidation exhibited by the latter (E_0_(**4**) = 1.35 and E_0_(Br^•/−^) = 1.92 V vs. NHE). The determination of the total quantum yield for bromide photo-oxidation, defined as $\Phi_{total}=\Phi_{inj}\times {\;\Phi }_{reg}$, among the series was on the basis that **34** had the highest $\Phi_{total}$ of 0.64. Photoelectrochemical experiments were conducted in a standard three-electrode cell by separating the anode and cathode compartments with a PEM membrane, in the presence of 1 M HBr solution. **33** showed the best performance among the series (Figure 24b), with a sustained photocurrent close to 1.5 mA cm^−2^, an IPCE of 11%, and an FE (%) of 81%. The authors noted that the most efficient compounds were not the most oxidant or photoreductant among the series but, rather, they exhibited a balance between both the properties. **33** displayed sufficient oxidation ability for near-unity regeneration and it was also photoreductant enough to enable a significant electron injection. 

Other promising results towards the HBr splitting were obtained in the presence of Zinc porphyrin-based sensitizers. Orbelli et al. [75] reported a series of perfluorinated Zn^II^ porphyrins bearing different acceptor linker groups, investigating the electronic and photoelectrochemical properties in the framework of DSPECs, to drive the HBr and water splitting. The authors introduced four electron-withdrawing C_6_F_6_ groups to obtain an electron-deficient tetrapyrrolic ring in order to increase the driving force for electron transfer in oxidation reactions. They also introduced a π-conjugated electron acceptor linker equipped with either a carboxylic or a cyanoacrylic group (see the structures of **15**, **16,** and **17** in Table 1), tasked to bind metal oxides and drive an efficient directionality of the excited state. The absorption spectra in the toluene medium of **15** and **16** reported the typical pattern of β-substituted Zn^II^ porphyrins, showing two well-defined Q bands and a prominent redshift of about 10 nm if compared to the meso-based porphyrin (**17**). This is a consequence of the more pronounced push–pull character of the formers **15** and **16** dyes (Figure 25a). The nanosecond laser spectroscopy was employed to gain insight into the electron dynamics occurring onto TiO_2_ or SnO_2_/TiO_2_ (active layer of SnO_2_ with a thin compact TiO_2_ overlayer) electrodes. The relatively negative dye excited state oxidation potential (E_ox_^*^ ≈ −0.66 V vs. NHE) should lead to a significant charge injection also onto TiO_2_ but the authors revealed an excited state residual on the ns–µs timescale. This was ascribed to the introduction of the perfluorinated groups which may significantly increase the inner-sphere reorganization energy (λ_in_) (it is estimated to be close to 0.8–1 eV), affecting the activation energy for electron transfer ΔG_ET_^*^, according to the Marcus equation ΔG_ET_^*^ = (E^*^_ox_ − E_FB_ + λ)^2^/4λ) where λ = λ_in_ + λo (λo is the outer-sphere reorganization energy) [159]. Due to the lower flat band potential of the SnO_2_/TiO_2_ system (ca. 0.44 V vs. NHE), this resulted in a higher injection quantum yield. SnO_2_ was first introduced by Bergeron et al. [160] for dyes that are not capable of an efficient charge injection owing to their lower reductant excited state and then, it was introduced in the framework of DSPECs by other authors. In both conditions, the charge-separated state (CS) was found to be long-lived: the recovery of the oxidized chromophore was far from being complete on a 50 µs time scale, allowing sufficient time for intercepting the target molecule of interest (Figure 25b). 

When Zn^II^-based porphyrins were loaded onto SnO_2_/TiO_2_, the photocurrent density–voltage curve in 0.1 M HBr underwent a 5- to 10-fold increase in photocurrent density, suggesting an improved charge injection and a slowed recombination, likely attributable to the efficient surface state passivation by TiO_2_. An optimized electrolyte mixture was found with 0.3 M NaBr and 0.1 M HBr, resulting in the highest photocurrent of 0.4 mA cm^−2^ for **16** (Figure 26). Under these conditions, the IPCE% was 7% and 2% in correspondence of the B and Q features, respectively. Combining the APCE (=Φ_inj_η_reg_) data with the regeneration efficiency computed from nanosecond laser kinetics in the presence of Br^-^, an injection quantum yield of only 14% was obtained.

Later, some of the authors reported on a new family of two asymmetric perfluorinated Zn^II^ porphyrins modified with an electron withdrawing branch in either β (**36**) or *meso* (**37**) positions for HBr splitting [76]. Additionally, **36** was further modified by introducing benzothiadiazole (BTD) moieties, aimed at enhancing the absorption at longer wavelengths. DFT and TD-DFT calculations revealed that delocalization of the excited states on the electron withdrawing link was partially successful with the BTD electron acceptor (Figure 27). However, the electronic density on the cyano-acrylic moiety remained comparatively smaller than to that localized in the ring. The photocurrent extrapolated for **36** resulted in higher results than those one of **37**, mainly due to the higher excited state directionality. However, the photocurrent associated with the Br^−^ oxidation resulted relatively low, especially if compared to the photocurrent obtained from the oxidation of the ascorbic acid (AA) as a hole scavenger, which reached the value of 300 µA cm^−2^. Analysis of the transient absorption spectrum by means of the nanosecond laser apparatus indicated that the 10-fold decrease in photocurrent was mainly ascribed to the insufficient driving force for the HBr splitting, paving the way for the design and synthesis of efficient sensitizers.

All the photoelectrochemical details such as the type of sensitizer, the photocurrent density, and the applied potential as well as the experimental conditions are summarized in Table 6.

**Table 5 molecules-29-00293-t005:** Structures of the sensitizers reported in the main text to drive HBr splitting.

Structure				
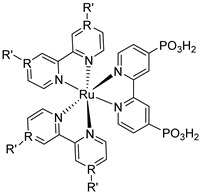	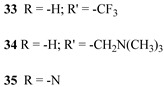	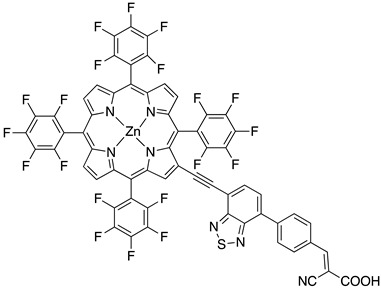		36
S [154,158]		S [76]		
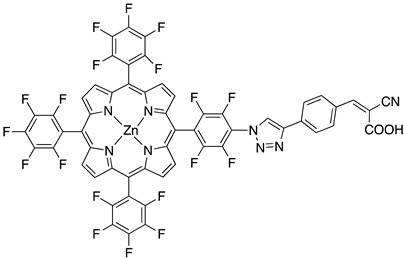			37	
S [76]				

## 7. Organic Reactions

Nowadays, DSPECs have been extensively explored in the field of organic reactions, aiming to replace expensive noble metal-based catalysts and achieve valuable chemical transformations, such as the generation of new carbonyl compounds or biomass remediation. This approach allows for the efficient replacement of the oxygen evolution reaction (OER) with an alternative oxidative pathway, coupled with the production of storable fuels like hydrogen (H_2_) or methane (CH_4_). After preliminary contributions concerning the photodehydrogenation of isopropanol to acetone [161] and to other photocatalytic systems [162,163,164], in 2014, T. Meyer [97] reported the photo-driven oxidation of benzyl alcohol to benzaldehyde in the presence of **7** (see Table 1), which was employed before as a WOC for water oxidation [165]. The catalyst was co-loaded onto a *nano*ITO/TiO_2_ core/shell system with the chromophore **4** (see Table 1). Under the presence of 0.1 M of benzyl alcohol, the system was able to generate current densities of 60 and 200 µA cm^−2^ for TiO_2_ and the core/shell system, respectively. After other attempts to successfully perform the alcohol oxidation by coupling both a ruthenium chromophore and a Ru-based catalyst, the tendency was to start to replace this expensive and toxic noble metal with fully organic chromophores or catalysts. A comprehensive historical overview of this progress can be found in a recent review [166]. One of the most promising approaches involves the use of the low-cost and readily accessible, 2,2,6,6-tetramethyl-1-piperidine N-oxyl (TEMPO), a well-known organocatalyst for selective and clean electrochemical alcohol oxidation [167,168,169,170]. In this contest, the pristine form of the catalyst TEMPO can be easily restored by reacting with the target substrate, as depicted in the reaction mechanism shown in Figure 28 [171]. TEMPO has been employed in electrocatalysis for selective oxidation reactions [172] and, more recently, it has been applied in photoelectrocatalysis by Choi [173] and Berlinguette [174] to promote the 5-hydroxymethylfurfural (HMF) at a BiVO_4_ semiconductor. In addition, due to its fast electrochemistry, TEMPO has also been used as a redox mediator for dye-sensitized solar cells (DSSCs) [86,175].

The group of Odobel reported an example of a DSPEC based on a triad (NDADI-Ru(bpy)_3_-TEMPO, where NADI = naphthalenedicarboxyanhydride) (**38**, see Table 7) was supported onto ITO nanoparticles for *para*-methoxybenzyl alcohol (Meo-BA) oxidation to para-methoxybenzaldehyde (Figure 29a) [176]. 

The dyad NDADI-Ru(bpy)_3_ was designed to enhance the charge separation, leading to the formation of the *ITO(e^−^)NDADI/Ru(III) CS*, as previously reported by Wenger and co-workers [177]. The TEMPO has been incorporated to minimize the spatial separation from the chromophore. After having found suitable conditions for TEMPO-mediated benzyl-alcohol oxidation, the authors employed the supported triad to perform the photo-driven reaction. The chopped photocurrent–density voltage curve (Figure 29b), in the presence of 50 mM of MeO-BA, demonstrated an increased photocurrent response for applied potentials higher than 0.2 V, resulting in a net photocurrent of about 150–180 µA cm^−2^. Under optimal conditions, the average Faradaic efficiency was on the order of 80%. To explore the replacement of the rare, toxic, and expensive ruthenium-based complexes with zinc porphyrins, to perform the same target reaction, Odobel’s group synthesized two porphyrin dyads covalently linked to TEMPO, bearing a carboxylic acid (**39**) or hydroxamic acid (**40**) moiety, respectively. Their structures are displayed in Table 7. The hydroxamic acid anchoring group of **40** was expected to provide higher stability of the TiO_2_-dye bond over time.

The photocurrent analysis, at pH 8 for **40**, highlighted the important role played by the TEMPO catalyst in performing the alcohol oxidation. As depicted in Figure 30, for a voltage larger than −0.06 V vs. NHE, the photocurrent density reached the limit value of 150–200 µA cm^−2^, significantly higher than the experiment performed without the catalyst (<<50 µA cm^−2^). The authors also explored the alcohol oxidation in acetonitrile medium because this organic solvent is compatible with proton reduction and is even more interesting for CO_2_ reduction owing to its 7–8 times higher solubility in organic solvents than in water [178]. In the case of the carbon dioxide reduction, this result is interesting both with respect to the possibility of having a higher substrate concentration at the cathodic counterpart and of avoiding CO_2_ hydration products, affecting the thermodynamic potentials for generating certain reduction molecules. In these conditions, a base-rich electrolyte is required as the oxidation resulted in proton release (Figure 28). The lower efficiency observed in acetonitrile medium was ascribed to the faster desorption rate of the dyad owing to their higher solubility in organic medium. In all the situations, the Faradaic efficiency for the aldehyde production ranged from 76 to 93%. Comparable experiments have been performed with **39**; this device gave a similar photocurrent and FE (%) in water while the faster dye desorption in acetonitrile resulted in impossibility to quantify the produced aldehyde, confirming the higher stability provided by the hydroxamic acid moiety. 

DSPECs have recently found applications in exploring biomass valorization via oxidative remediation. For instance, the glycerol oxidation products find application in cosmetics and in the preparation of polyesters and adhesives [179]. Generally, DSPECs are not suitable for this molecular conversion owing to the basic conditions required (pH 8.5), which result in dye desorption [180,181]. In this contest, to drive the glycerol oxidation mediated by TEMPO, Bruggeman et al. reported on a thienopyrroledione-based dye (**41**, Table 7) encased in an acetonitrile-based redox-gel, which is able to protect the TiO_2_ photoanode from degradation [182]. The DPSEC was assembled by generating a by-phasic system in which the **41-**based photoanode was embedded in a poly (vinylidene fluoride*-co-*hexafluoropropylene (PVDF-HFP) gel layer and the resulting assembly was put in contact with an aqueous electrolyte. This arrangement allows TEMPO to regenerate the photo-oxidized dye which, in turn, produces the oxammonium TEMPO(+) cation that can diffuse through the gel to reach the gel–aqueous interface where the glycerol oxidation takes place, as depicted in Figure 31a.

The FE (%) computed by analyzing the products accumulated by chronoamperometry experiments (Figure 31b) of the full water- and the TEMPO redox-gel-based systems, resulted in almost quantitative results for both cases. However, the latter device exhibited at least 10-fold higher photocurrent density, thereby leading to faster overall conversion. In addition, improved stability was outlined by devices assembled with the redox-gel system, which preserved its typical orange color throughout the entire 48 h electrolysis. This confirmed how the use of the PVDF-HFP gel layer was effective in preserving not only the chemical stability of the sensitizer but, in general, that of the entire photoanode unit. By contrast, in the same experimental conditions, devices fabricated with the conventional architecture no longer generated photocurrent after 15 h as a consequence of the dye desorption. Thus, albeit the two systems exhibited comparable FE (%) values, in the presence of the redox gel the overall yield of glycerol oxidation was larger if compared to the full water architecture. It is worth noting that just a slight percentage of TEMPO (~6%) was found in the aqueous layer after 48 h, confirming that the mediator is strongly incorporated into the redox gel matrix, which in fact avoids the diffusion of TEMPO outside the film, keeping it in a region close to the semiconductor interface. Moreover, only a slight amount of water (~1%) was found entrapped into the redox-gel after the same period of time (48 h), which is beneficial to give effective protection to the photoanode, thereby preventing the possible hydrolysis of the Ti-Dye bond. The by-phasic architecture of these devices highlights a smart way for stabilizing the dyes towards alkaline environments without the employment of expensive and demanding techniques, such as the aforementioned ALD [59]. Apart from glycerol oxidation, in the framework of biomass remediation, the photo-assisted lignin oxidation, or cleavage, is another promising application in which, to date, only a few examples linked to DSPECs have been reported. Lignin represents the largest noncarbohydrate component in lignocellulosic biomass and its native structure is characterized by interunit linkages including alkyl-aryl-ether, phenylcoumaran, resinol, biphenyl, and other related groups [183]. In 2020, G. Leem reported a photochemical device able to drive the initial oxidation of the secondary benzyl alcohol in lignin model compound (LMC, characterized by the most abundant interunit linkage in natural lignin) and real lignin, as part of a selective two-step process prior to C–O bond cleavage in the lignin backbone [184]. Furthermore, the same group reported a heterogeneous approach to cleave the C*_aryl_*-C*_α_* bond in a phenolic LMC by employing a one-dimensional TiO_2_ nanoroad array (TiO_2_ NRAs) DSPEC in combination with a Halogen Atom Transfer (HAT) mediator [185]. **1** (Table 1) was employed as a sensitizer, while 4-acetamido-2,2,6,6tetramethyl-1-peperidine-N-oxyl (ACT) was used as the HAT system (Figure 32). 

The photocurrent response of the FTO|TiO_2_ NRAs|**1** has been investigated in acetonitrile by increasing the ACT concentration and by analyzing the chopped photocurrent density–time response. When the concentration of ACT was kept constant, the addition of the LMC led to a significant increase in the response from 190 to 690 μA cm^−2^, as shown in Figure 33a. The presence of the cleavage products has been investigated by means of both ^1^H and ^13^C NMR after 5 h continuous illumination. According to the literature, the author demonstrated no oxidative cleavage of the C-C bond for the nonphenolic lignin model in DSPEC nor by following a thermal catalytic oxidation in the presence of a vanadium catalyst and pyridine at 80 °C for 48 h (Figure 33b). The authors concluded that the presence of the phenoxy group in LMC is an essential moiety to lead to the C-C cleavage. 

In the near future, several efforts should be taken to explore the possibility of reaching the C-C cleavage in the native lignin compound.

All the photoelectrochemical details such as the type of sensitizer and catalyst, the photocurrent density, and the applied potential as well as the experimental conditions are summarized in Table 8.

**Table 7 molecules-29-00293-t007:** Structures of the molecules reported in the main text to drive organic reactions. S represents a sensitizer, while A represents an assembly.

Structures				
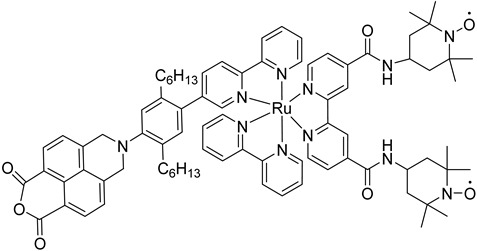			**38**	
A [176]				
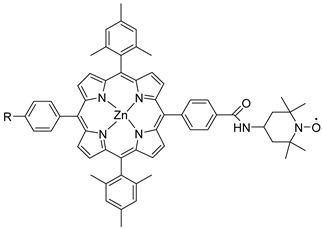	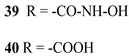	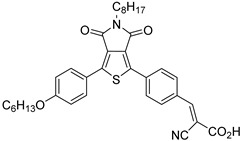		**41**
A [28]		S [182]		

## 8. Conclusions and Future Outlooks

Since Mallouk’s initial contribution in 2009, DSPECs have emerged as promising alternatives to the classical PEC devices, capable of converting solar light into chemical bonds such as O_2_, H_2_, or X_2_. Several efforts have been devoted to finding strategies to prevent the cleavage of the ester dye-TiO_2_ bond, especially under alkaline conditions. Several contributions reported higher stability for dye bearing –PO_3_H_2_ anchoring groups compared to their -COOH analog. However, both grafting units are precluded under alkaline conditions. Other promising grafting moieties, such as -py groups or diazonium groups, have been identified to enable stability under alkaline conditions. An ALD overlayer of Al_2_O_3_ or TiO_2_ has proven to be an efficient passivation strategy to prevent the hydrolysis of the ester bond. Since the outset of the DSPECs, the splitting of water has been considered the most interesting reaction due to its safety and natural abundance. Numerous sensitizers, catalysts, or assemblies of both have been developed. Alternative oxidative pathways have been explored to overcome limitations arising from using water. HX splitting results in a promising inexpensive storage way for solar energy, with particular attention given to HBr oxidation as a source of Br_2_ for fuel cells. Recently, organic reactions such as the generation of new carbonyl compounds have garnered attention in the scientific community as a strategy to replace the expensive and toxic noble metal employed until now. Future development should focus on identifying new stabilizing strategies to preserve dye absorption without affecting the electronic dynamics occurring at the semiconductor/dye interface, which could otherwise decrease the overall yield of the device. 

## Figures and Tables

**Figure 1 molecules-29-00293-f001:**
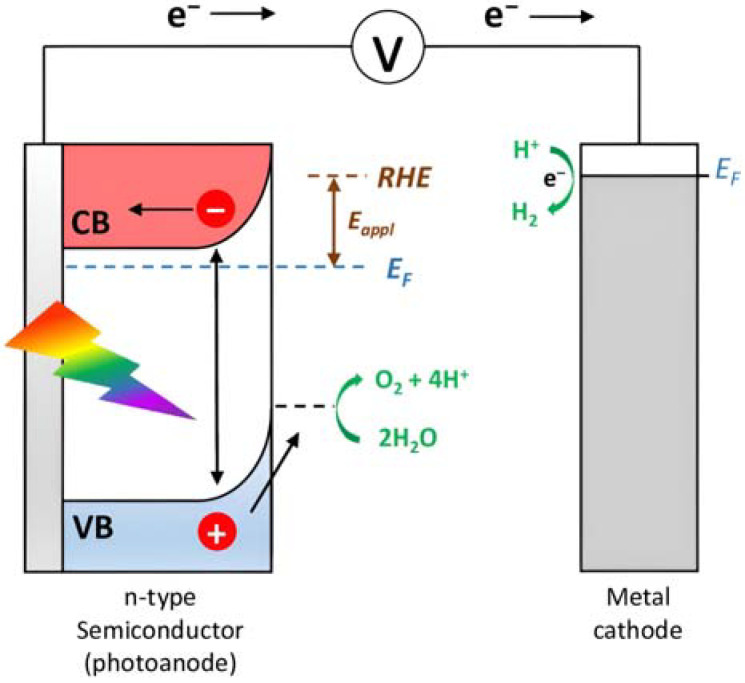
Schematic illustration of the basic functioning of a PEC.

**Figure 2 molecules-29-00293-f002:**
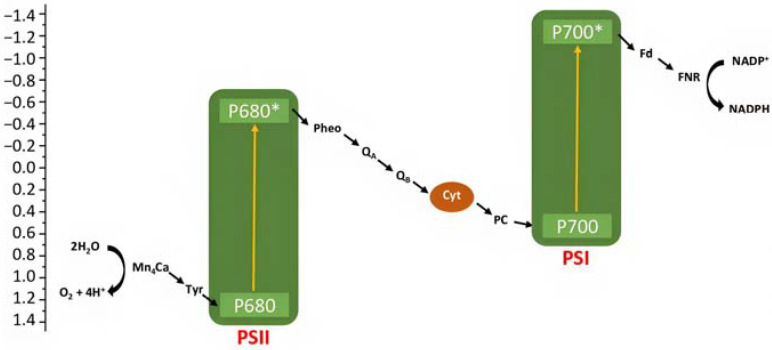
Simplified representation of the Z-scheme for a photosynthetic process.

**Figure 3 molecules-29-00293-f003:**
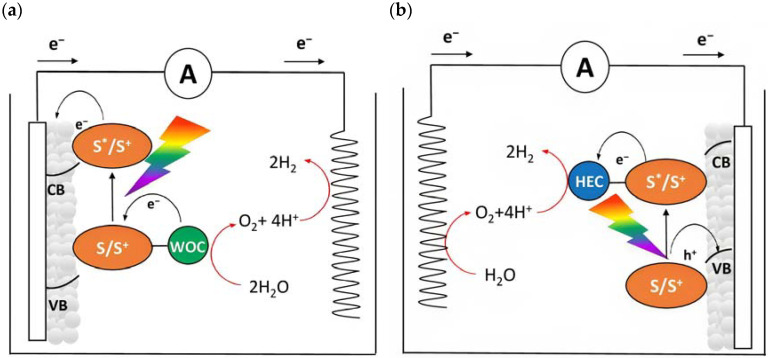
Schematic representation of a DSPEC in the framework of water splitting in an (**a**) n-type or in (**b**) p-type configuration in which the semiconductors, the sensitizers, the catalysts (WOC or HEC), and target reactions are depicted.

**Figure 4 molecules-29-00293-f004:**
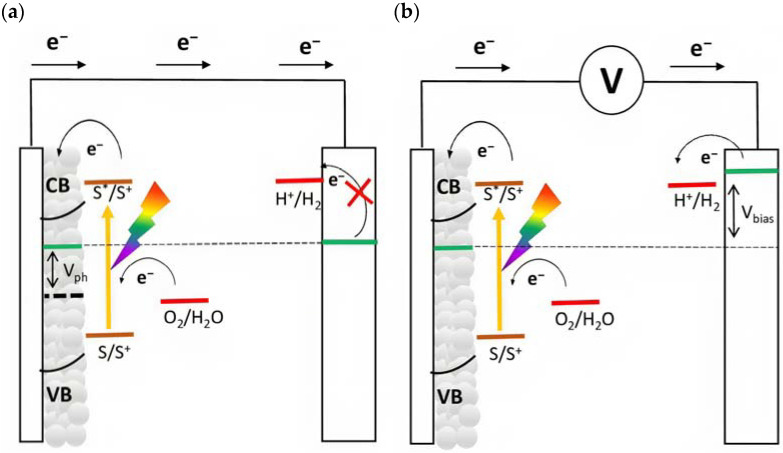
Schematic representation of energy levels for an n-type-based DSPEC for water splitting. The quasi-Fermi level (dotted line) of the n-type material is approximated with the flat band potential (V_fb_) of the semiconductor in condition of (**a**) zero bias and (**b**) under the application of a bias.

**Figure 5 molecules-29-00293-f005:**
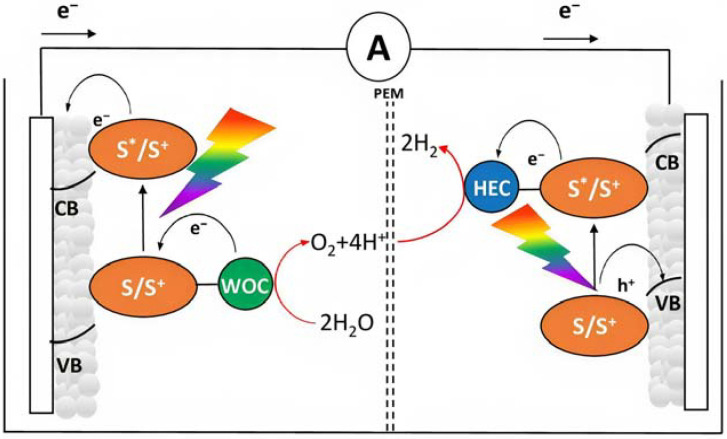
Tandem n-p DSPEC for solar water splitting. The photoanode (**left**) and photocathode (**right**) compartments are depicted.

**Figure 6 molecules-29-00293-f006:**
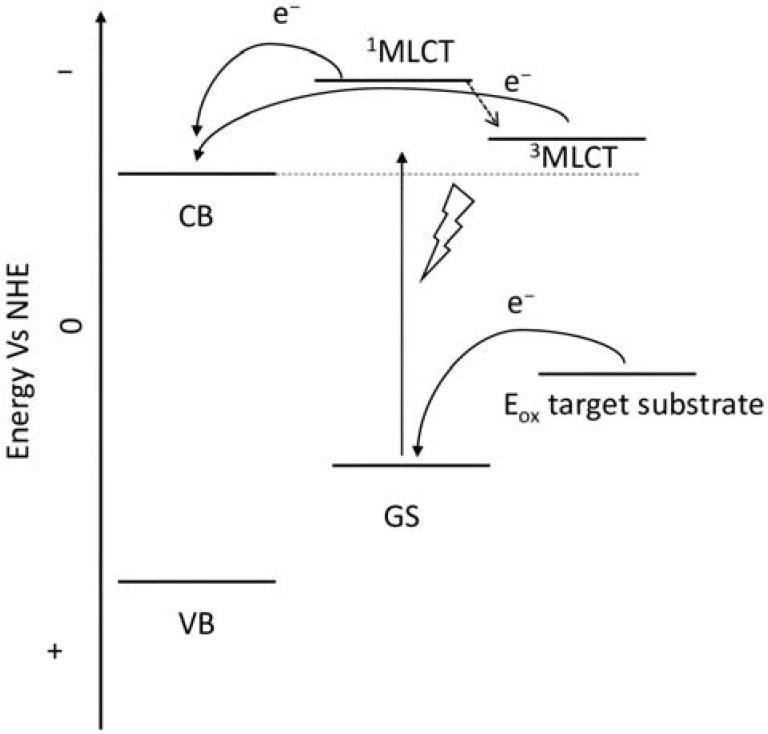
Representative scheme of the energetic levels of the semiconductor, a general ruthenium complex, and the target substrate.

**Figure 7 molecules-29-00293-f007:**
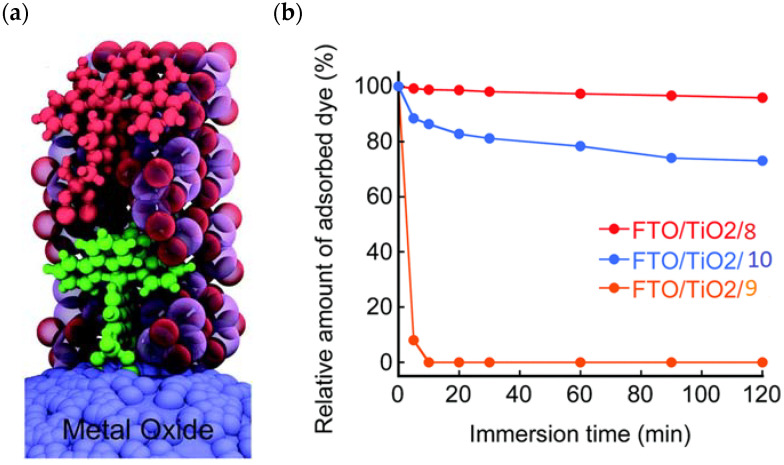
(**a**) Visualization of the ALD mummy-protected surface assembly with the dye **4** (green molecule) and the catalyst **7** (red molecule) embedded in ~3 nm of Al_2_O_3_ (purple and red spheres). (**b**) Plots of the relative amount of dye (**8**, **9**, and **10**) adsorbed on the TiO_2_ electrode as a function of immersion time in an aqueous acetate buffer solution (0.1 M, pH 5.0). Reprinted and adapted with permission of the Royal Chemical Society from refs. [60,61].

**Figure 8 molecules-29-00293-f008:**
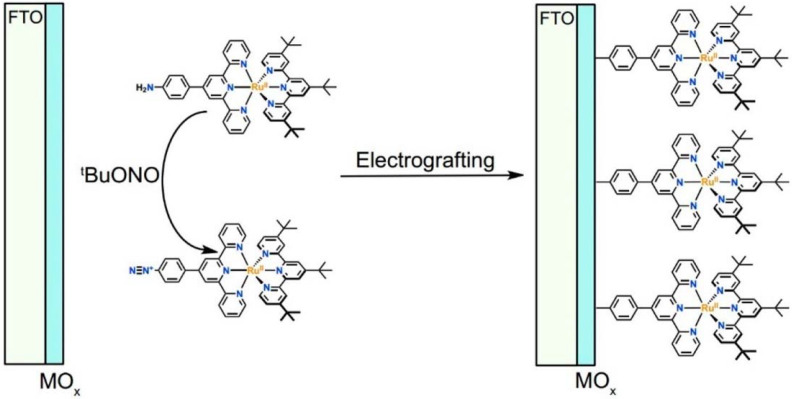
General strategy developed for electrografting diazonium-substituted sensitizer molecules on metal oxide (MOx) surfaces. Reprinted with permission of the American Chemical Society from ref. [69].

**Figure 9 molecules-29-00293-f009:**
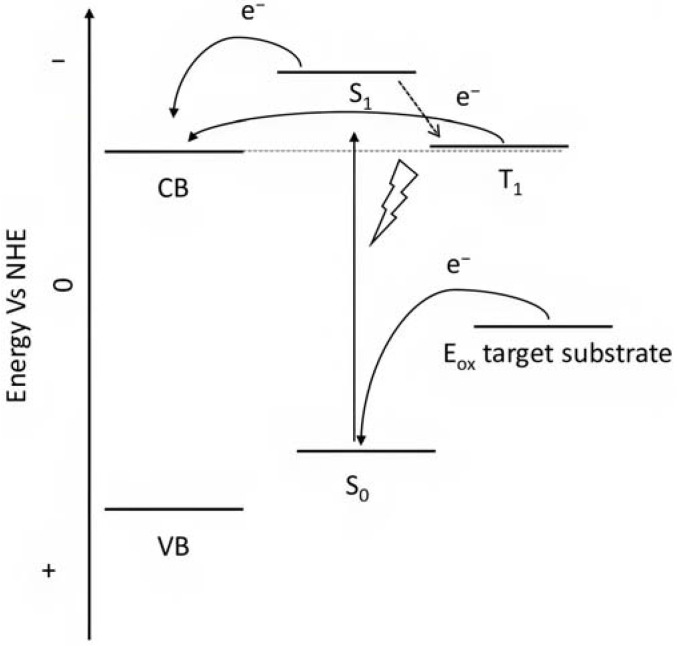
Representative scheme of the energetic levels of the semiconductor, a general porphyrin, and the target substrate.

**Figure 10 molecules-29-00293-f010:**
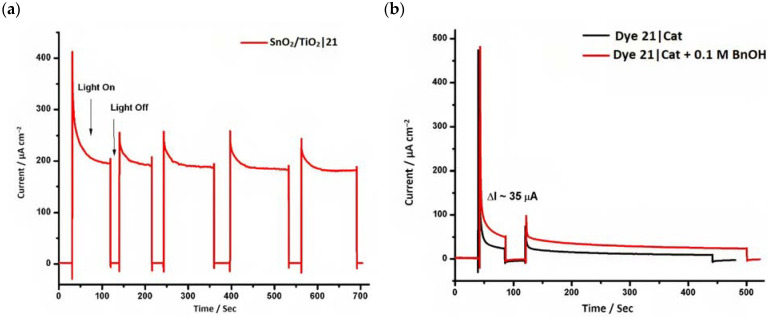
(**a**) Chopped current–time (I–t) profiles for SnO_2_/TiO_2_|**21** at pH 4.5, 0.1 M acetate buffer with 0.4 M LiClO_4_ and 20 mM H_2_Q, E_app_ = 0.2 V (**b**) Current–time (I–t) profile for **21** in 0.1 M acetate buffer with 0.4 M LiClO_4_ with and without 0.1 M BnOH. Reprinted and adapted with permission of the American Chemical Society from ref. [94].

**Figure 11 molecules-29-00293-f011:**
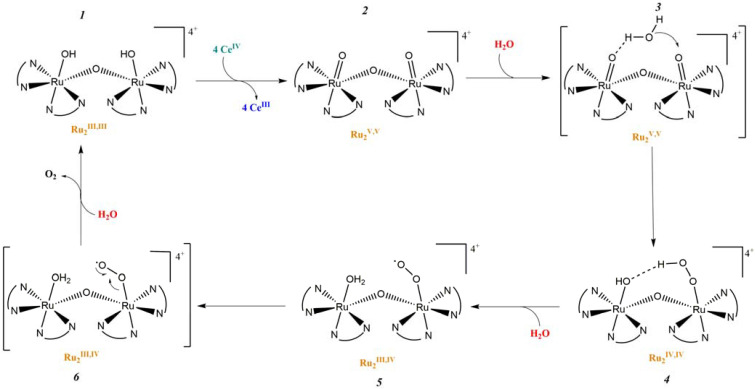
Proposed mechanism of oxygen evolution in μ-oxo bridged blue dimer-type Ru (II) complexes.

**Figure 12 molecules-29-00293-f012:**
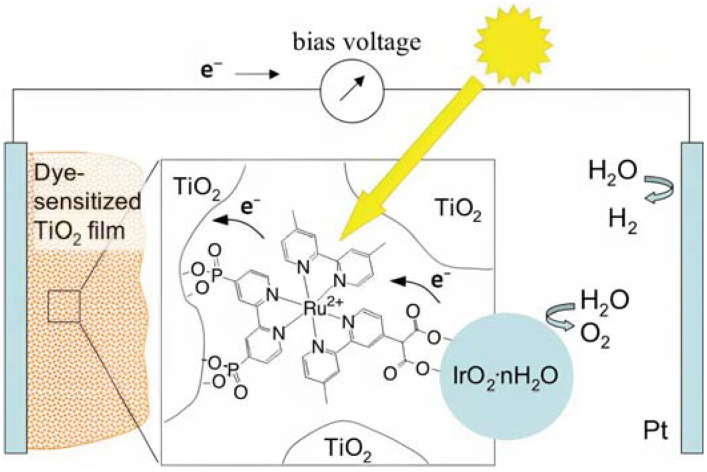
Schematic representation of the main components of the DSPEC, namely the TiO_2_ semiconductor, the sensitizer (**22**), and the IrO_2_·nH_2_O WOC. Reprinted and adapted with permission from the American Chemical Society from ref. [19].

**Figure 13 molecules-29-00293-f013:**
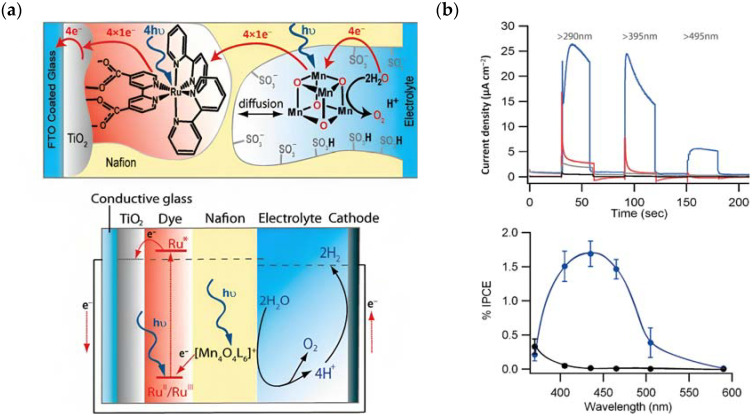
(**a**) Top: Schematic representation of the main component of the DSPEC (TiO_2_, the sensitizer (**1**) and the Mn-based WOC (**23**) and bottom: the main operating principles. (**b**) Top: chopped photocurrent density–time profile for **23**- Nafion/TiO_2_ (gray), Nafion/**1**-TiO_2_ (red), and **23**-Nafion/**1**-TiO_2_ (blue), illuminated at 100 mW cm^−2^ through a series of long pass light filters, and bottom: average incident photon to current efficiency (IPCE) plot for **23**-Nafion/**1**-TiO_2_ (blue) and TiO_2_ (black) electrodes. Reprinted and adapted with permission of American Chemical Science from ref. [95].

**Figure 14 molecules-29-00293-f014:**
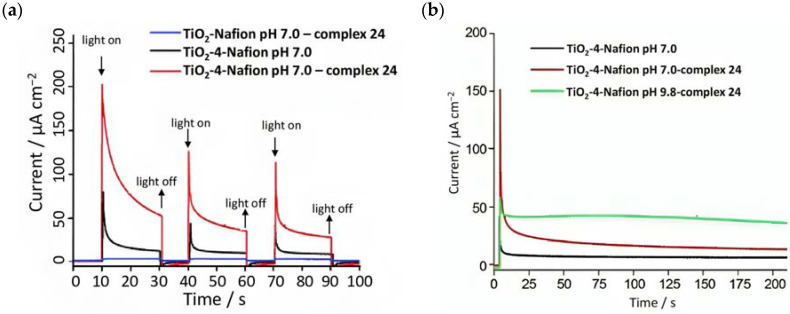
(**a**) Chopped photocurrent density–time profile for the main components and the whole system and (**b**) photocurrent decays under different pH conditions. Reprinted and adapted with permission of the Royal Chemical Society from ref. [96].

**Figure 15 molecules-29-00293-f015:**
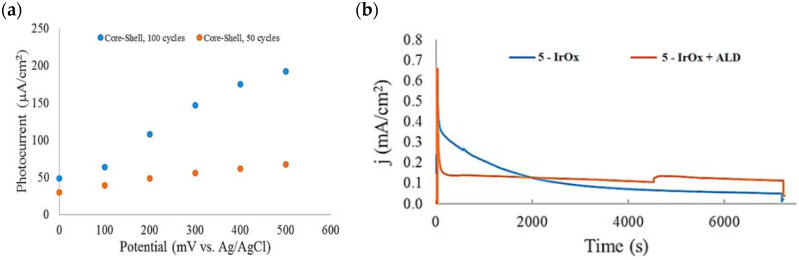
(**a**) Photocurrent density–voltage profile of the **5**–IrO_X_ NP assembly on two different core/shell electrodes (50 (red) and 100 (blue) cycles) at pH 5.8. (**b**) Photocurrent density–time curves for the *nano*ITO/TiO_2_ core/shell-based electrodes with and without the ALD overlayer. Reprinted and adapted with permission from the American Chemical Society from ref. [98].

**Figure 16 molecules-29-00293-f016:**
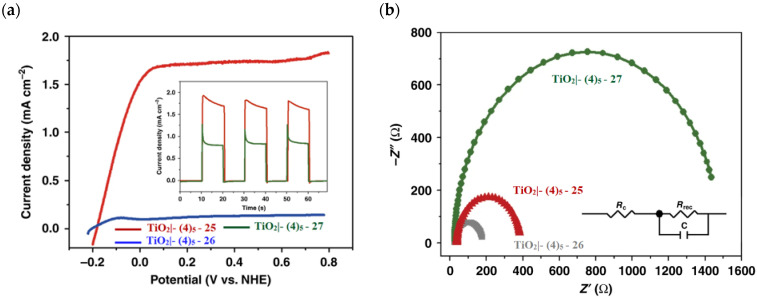
(**a**) Photocurrent density–voltage curves of TiO_2_|-(**4**)_5_,**25** and TiO_2_|-(**4**)_5_,**26** in a 0.1 M acetate buffer/0.5 M NaClO_4_. Insert: chopped photocurrent density–time curves for TiO_2_|-(**4**)_5_,**25** and TiO_2_|-(**4**)_5_,**27** at a constant bias of 0.2 V vs. NHE. (**b**) Nyquist plot for TiO_2_|-(**4**)_5_,**25**, TiO_2_|-(**4**)_5_,**26**, and TiO_2_|-(**4**)_5_,**27** with the equivalent circuit used for data fitting. Reprinted and adapted with permission of Nature from ref. [99].

**Figure 17 molecules-29-00293-f017:**
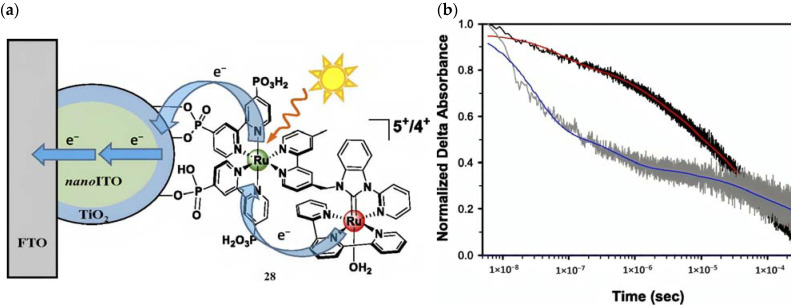
(**a**) Schematic representation of the FTO|*nano*ITO/TiO_2_-[Ru_a_^II^-Ru_b_^II^-OH_2_]^4+^ assembly and (**b**) 450 nm decays for the chromophore- (dark gray) and **28**- (black) derivatized FTO|*nano*ITO/TiO_2_ in 0.5 M LiClO_4_/10 mM acetate buffer pH 4.6. Reprinted and adapted with permission of PNAS from ref. [130].

**Figure 18 molecules-29-00293-f018:**
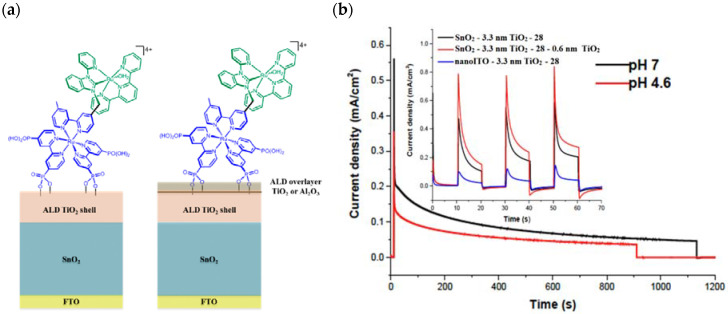
(**a**) Representation of an ALD core/shell electrode surface with and without ALD overlayer. (**b**) Photocurrent–time curves for FTO|SnO_2_/TiO_2_(6.6 nm)|-[Ru^II^_a_-Ru^II^_b_-OH_2_]^4+^ with the TiO_2_ overlayer in 0.5 M LiClO_4_/20 mM acetate buffer at pH 4.6 (red) and in a 0.1 M phosphate buffer at pH 7/0.5 M NaClO_4_ (black). Insert: chopped photocurrent density–time curves comparisons between SnO_2_ and *nano*ITO core/TiO_2_ photoanodes, FTO|SnO_2_/TiO_2_|-[Ru^II^_a_-Ru^II^_b_-OH_2_]^4+^ (black), and FTO|nanoITO/TiO_2_|-[Ru^II^_a_-Ru^II^_b_-OH_2_]^4+^ (blue) with ALD TiO_2_ shells (3.3 nm) in 0.5 M LiClO_4_/20 mM acetate buffer. The red curve shows the impact of the TiO_2_ overlayer on the photocurrent profile of the SnO_2_-based electrode. Reprinted and adapted with permission of PNAS from ref. [131].

**Figure 19 molecules-29-00293-f019:**
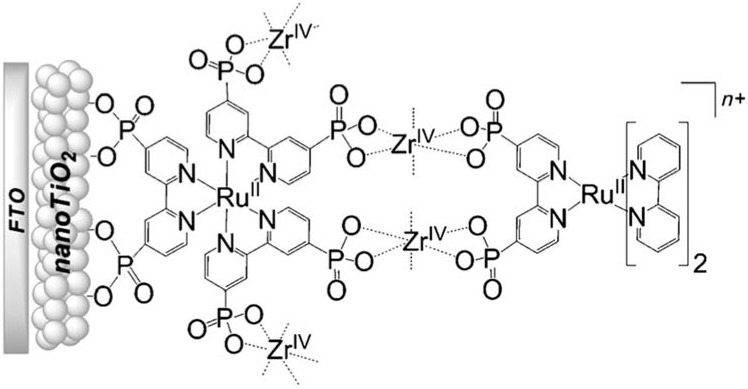
Schematic representation of the bi-layer film assembly. Reprinted with permission of Wiley from ref. [136].

**Figure 20 molecules-29-00293-f020:**
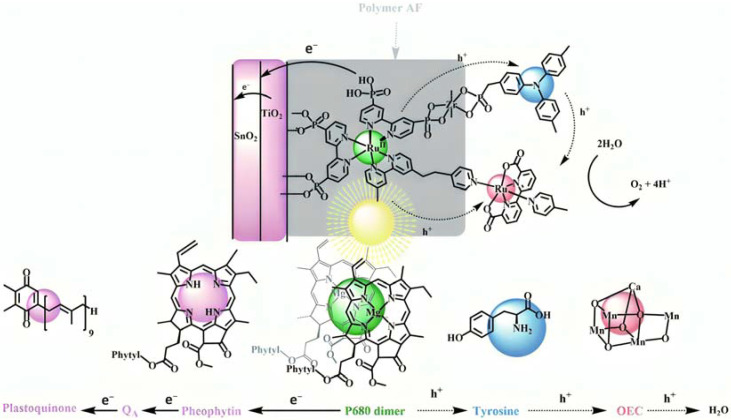
Illustrative representation of the DSPEC and its electron flow for a fully assembled system and their analog in the natural PSII. Reprinted and adapted with permission of the Royal Chemical Society of Chemistry from ref. [139].

**Figure 21 molecules-29-00293-f021:**
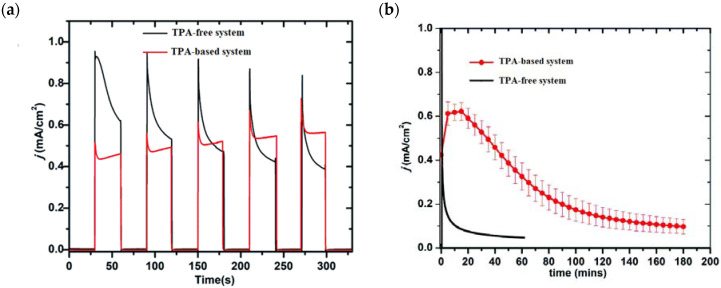
(**a**) Chopped photocurrent density–time (j–t) curves for TPA-free (black) TPA-based (red) systems at an applied bias of 0.6 V vs. NHE in 0.1 M phosphonate buffers/0.4 M NaClO_4_ pH 7.0. (**b**) Electrolysis of 3 h duration of the two systems described in (**a**). Reprinted and adapted with permission of the Royal Chemical Society of Chemistry from ref. [139].

**Figure 22 molecules-29-00293-f022:**
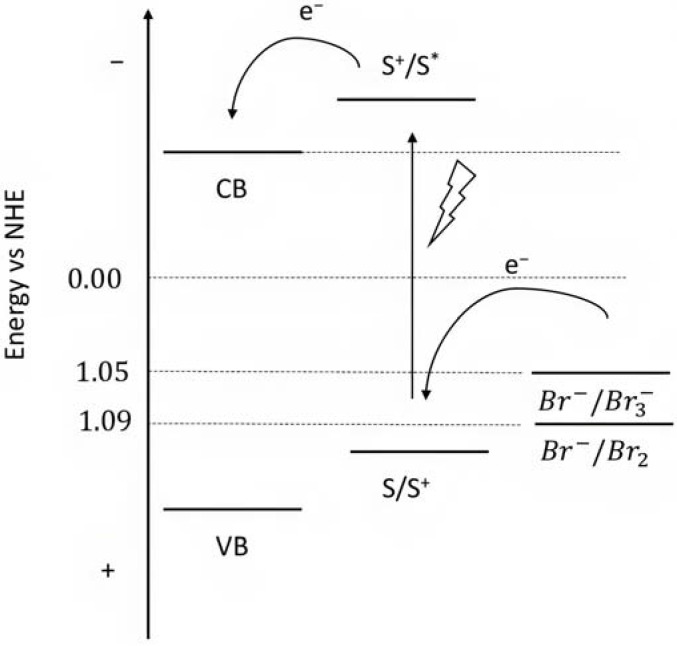
Representative scheme of the energetics levels for the Br^−^ photooxidation.

**Figure 23 molecules-29-00293-f023:**
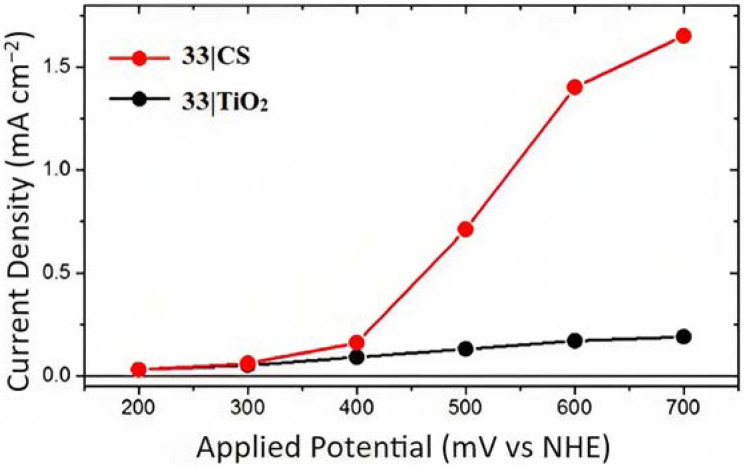
Photocurrent density–voltage profile for **33**|CS (red) and **33**|TiO_2_ (black). Reprinted and adapted with permission of the American Chemical Society from ref. [154].

**Figure 24 molecules-29-00293-f024:**
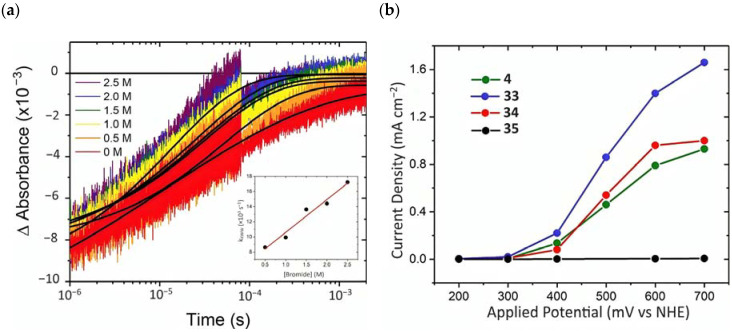
(**a**) Kinetics of 405 nm recorded for **34**|CS in 1 M HClO_4_(aq) (argon saturated) with the indicated bromide concentrations. In black is the reported fitting with the KWW function. (**b**) Photocurrent density–voltage curves for the different complexes (**4**, **33**, **34,** and **35**) in 1 M HBr under white illumination. Reprinted and adapted with permission of the American Chemical Society from ref. [158].

**Figure 25 molecules-29-00293-f025:**
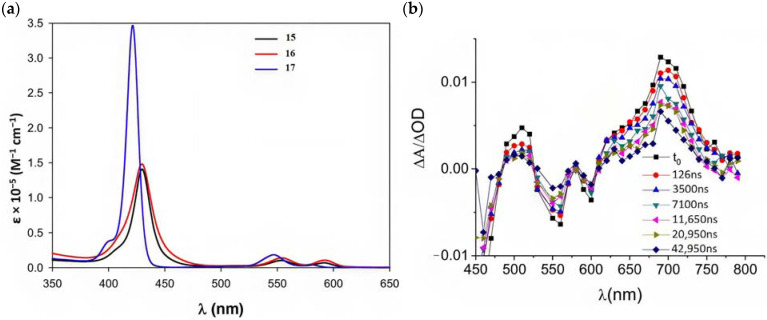
(**a**) Absorption spectra recorded for **15**, **16,** and **17**. (**b**) Transient absorption spectra for **16** loaded on SnO_2_/TiO_2_. Thin films were in contact with 0.1 M NaClO_4_, pH 3. Reprinted and adapted with permission of the American Chemical Society from ref. [75].

**Figure 26 molecules-29-00293-f026:**
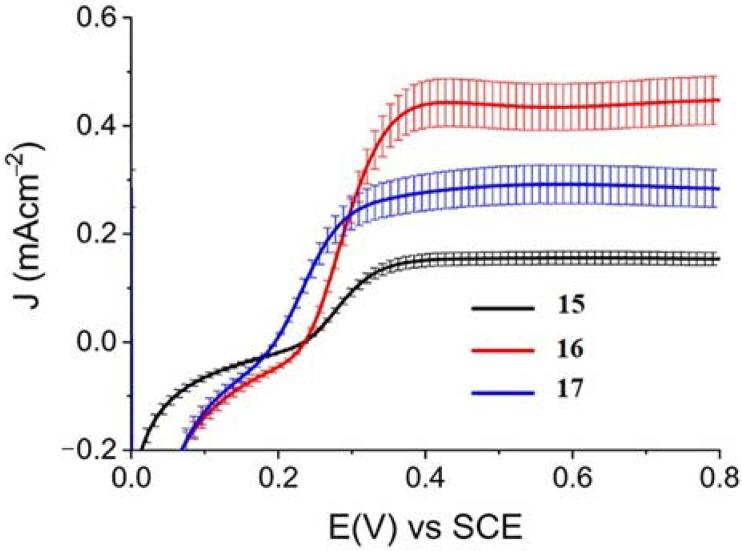
Photocurrent density–voltage curves for **15**, **16,** and **17** in contact with 0.1 M HBr/0.3 M NaBr. Reprinted and adapted with permission of the American Chemical Society from ref. [75].

**Figure 27 molecules-29-00293-f027:**
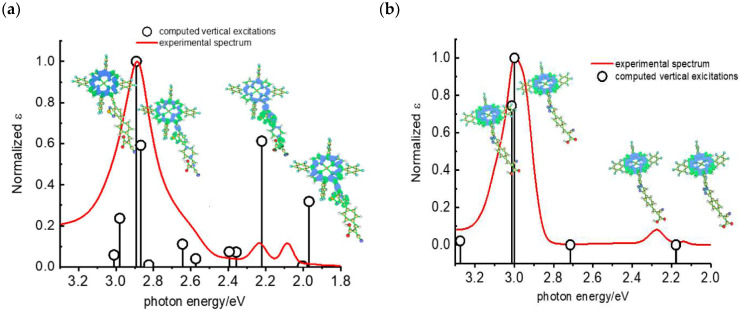
Computed vertical transitions at the TDDFT 6311 Gd,p level (black vertical bars) overlapped with the experimental spectrum of (**a**) **36** and (**b**) **37**. Reprinted and adapted with permission of MDPI from ref. [76].

**Figure 28 molecules-29-00293-f028:**
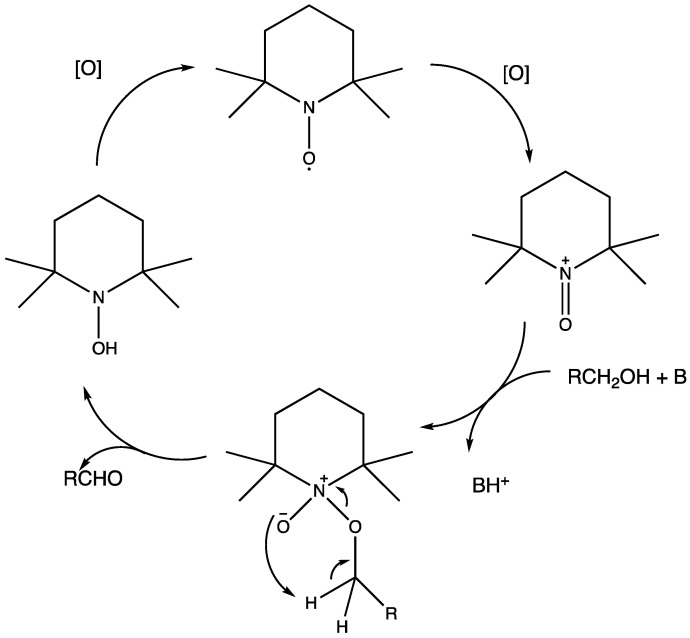
Structure of TEMPO mediator together with the oxidative cycle involved in the alcohol oxidation in the presence of a base.

**Figure 29 molecules-29-00293-f029:**
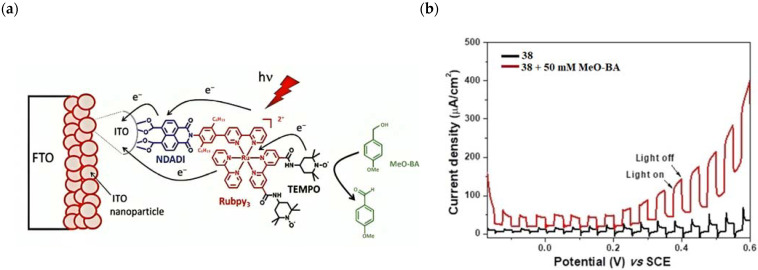
(**a**) Schematic representation of the photoanodic compartment for **38** triad supported onto ITO nanoparticles. (**b**) Photocurrent density–voltage curves under chopped light irradiation of triad **38** in the absence (black trace) and in the presence of 50 mM of MeO-BA (red trace) in carbonate buffer at pH 10. Reprinted and adapted with permission of Wiley from ref. [176].

**Figure 30 molecules-29-00293-f030:**
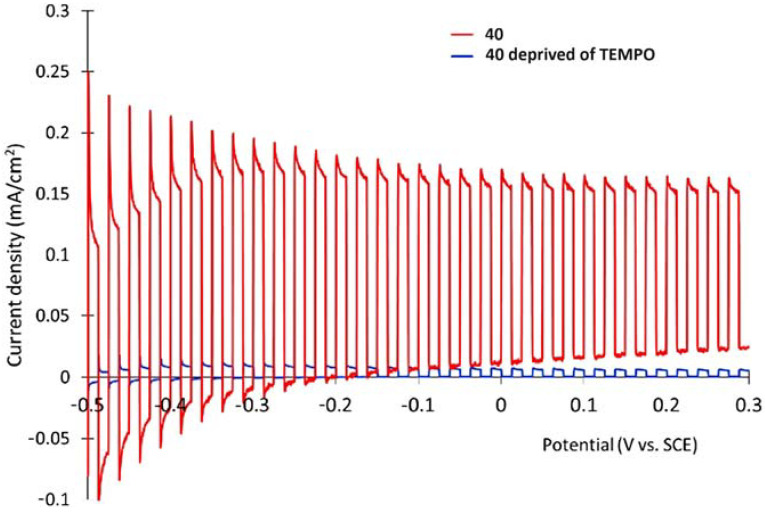
Chopped light photocurrent–density voltage curves recorded under white light irradiation for **40** and **40** deprived of TEMPO on a TiO_2_ electrode in contact with a 0.1 M borate buffer at pH 8, containing 50 mM [MeO-Ph-CH_2_OH] and 0.1 M [NaClO_4_]. Reprinted and adapted with permission of the American Chemical Society from ref. [28].

**Figure 31 molecules-29-00293-f031:**
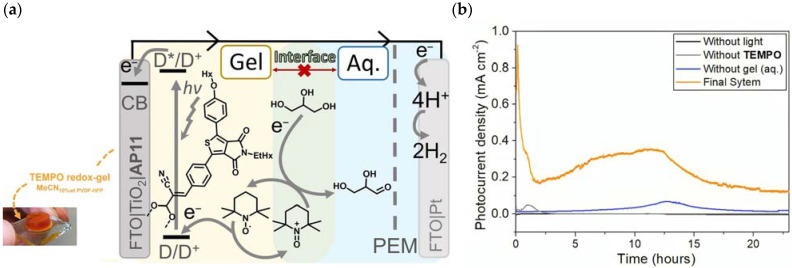
(**a**) Schematic representation of an aqueous (blue) biphasic DSPEC with a TEMPO containing redox-gel layer (yellow) for glycerol oxidation. (**b**) Photocurrent density–time curve in light in the absence of TEMPO, in the full water condition, for the final by-phasic device and in the dark. Reprinted and adapted with permission of Wiley from ref. [182].

**Figure 32 molecules-29-00293-f032:**
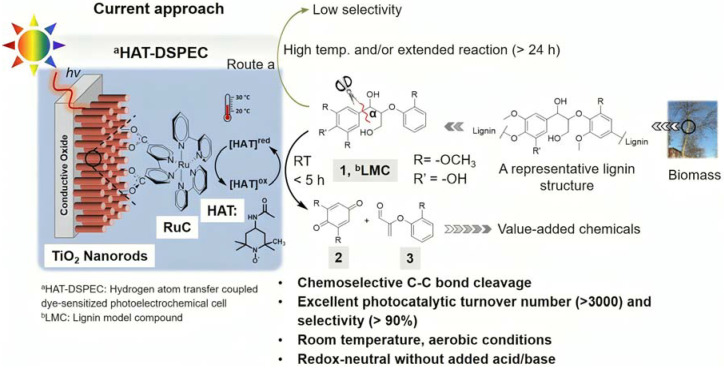
Illustration of the FTO|TiO_2_ NRAs|**1** in the presence of the HAT and the LMC. Reprinted and adapted with permission of the American Chemical Society from ref. [185].

**Figure 33 molecules-29-00293-f033:**
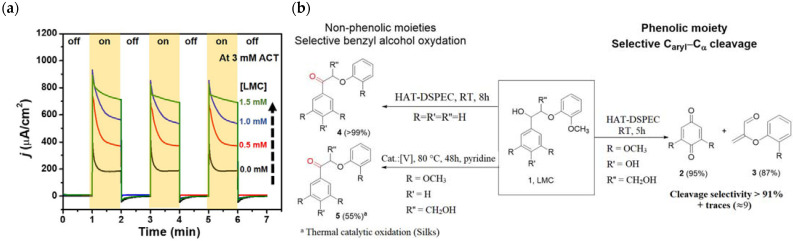
(**a**) Photocurrent density-time curves for FTO|TiO_2_ NRAs|**1** electrodes in the presence of 3 mM ACT and increasing LMC concentrations from 0 to 1.5 mM. (**b**) Different oxidation/cleavage pathways for the LMC in the presence of different substituents and conditions. Reprinted and adapted with permission of the American Chemical Society from ref. [185].

**Table 2 molecules-29-00293-t002:** Historical overview of the main breakthroughs toward the realization of efficient Ru-based WOC.

Meyer 1982	Liobet 2004	Thummel 2005	Meyer 2008
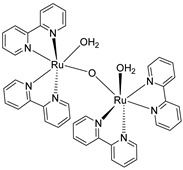	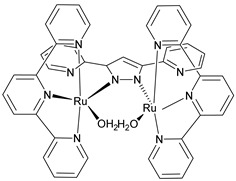	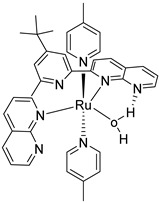	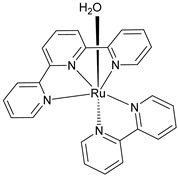
**I**	**II**	**III**	**IV**
**Sun 2009**	**Sun 2010**		**Sun 2009**
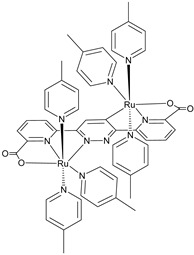	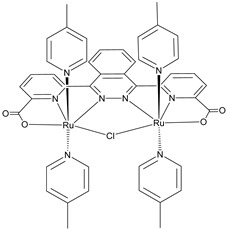		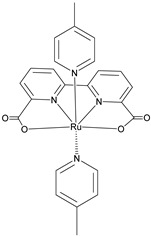
**V**	**VI**		**VII**

**Table 4 molecules-29-00293-t004:** Experimental details for the DSPECs described in the main text in the field of water splitting.

Sensitizer	Catalyst	Photocurrent	Details	Ref
TiO_2_@**22**	IrO_2_·nH_2_O	30 µA (>0.2 V vs. Ag/AgCl)	JV in 30 mM Na_2_SiF_6_/500 mM Na_2_SO_4_, pH 5.75	[19]
TiO_2_@**1**	**23**	15 µA (0 V)	Cronoamperometry in 0.1 M Na_2_SO_4_ pH 6.5	[95]
TiO_2_@**4**	**24**	50 µA (0 V)	Cronoamperometry in 0.1 M Na_2_SO_4_	[96]
*nano*ITO/TiO_2_@**5**	IrO_x_	110 µA cm^−2^ (0.3 V vs. Ag/AgCl)	Cronoamperometry in NaSiF_6_ pH 5.8	[98]
TiO_2_@**8**	/	1.1 mA cm^−2^ (−0.17 V vs. SCE)	Cronoamperometry in 0.1 M acetate buffer/30 mM EDTA/0.1 NaClO_4_ pH 5.0. Stable over 100 min	[61]
TiO_2_@**4**	**25** **26** **27**	1.5 mA cm^−2^ (0.2 V vs. NHE) 140 µA cm^−2^ (0.2 V vs. NHE) 0.8 mA cm^−2^ (0.2 V vs. NHE)	Chopped photocurrent density–time in a 0.1 M acetate buffer/0.5 M NaClO_4_	[99]
*nanoITO*–TiO_2_@**28** (assembly)		100 μA cm^−2^ (0.2 V vs. NHE)	Cronoamperometry in 20 mM acetate buffer pH 4.6	[130]
SnO_2_/TiO_2_@**28** (assembly)		100 μA cm^−2^ (0.6 V vs. NHE)	Cronoamperometry in 0.1 M phosphate buffer at pH 7/0.5 M NaClO_4_	[131]
TiO_2_@**30** (assembly)		500 μA cm^−2^ (0.2 V vs. NHE)	Chopped cronoamperometry in 0.1 M Na_2_SO_4_ pH 6.4	[137]
SnO_2_/TiO_2_@**5** SnO_2_/TiO_2_@**6**	**31**	0.97 mA cm^−2^ (0.1 V) 1.45 mA cm^−2^	Cronoamperometry in, 0.1 M acetate buffer/1.0 M in NaClO_4_ pH 5.7	[138]
SnO_2_/TiO_2_@**32**		600 μA cm^−2^ (0.6 V vs. NHE)	Cronoamperometry in 0.1 M phosphonate buffer/0.4 M NaClO_4_ pH 7.0	[139]

**Table 6 molecules-29-00293-t006:** Experimental details for the DSPECs described in the main text for the HBr oxidation.

Sensitizer	Photocurrent	Details	Ref
SnO_2_/TiO_2_@**33**	1.5 mA cm^−2^ (>0.7 V vs. NHE)	JV in 1 M HBr. FE% (4) = 72%	[154]
SnO_2_/TiO_2_@**4**	0.9 mA cm^−2^ (>0.7 V vs. NHE)	JV in 1 M HBr. FE% (4) = 51%, FE% (33) = 81%, FE% (34) = 70%, FE% (35) = 54%,	[158]
SnO_2_/TiO_2_@**33**	1.6 mA cm^−2^ (>0.7 V vs. NHE)		
SnO_2_/TiO_2_@**34**	1.0 mA cm^−2^ (>0.7 V vs. NHE)		
SnO_2_/TiO_2_@**35**	4.0 µA cm^−2^ (>0.7 V vs. NHE)		
SnO_2_/TiO_2_@**15**	100 μA cm^−2^ (>0.4 V vs. SCE)	JV in 0.3 M NaBr/0.1 M HBr	[75]
SnO_2_/TiO_2_@**16**	400 μA cm^−2^ (>0.4 V vs. SCE)		
SnO_2_/TiO_2_@**17**	300 μA cm^−2^ (>0.4 V vs. SCE)		
SnO_2_/TiO_2_@**36**	300 μA cm^−2^ (>0.4 V vs. SCE)	JV 0.1 M HBr	[76]
SnO_2_/TiO_2_@**37**	250 μA cm^−2^ (>0.4 V vs. SCE)		

**Table 8 molecules-29-00293-t008:** Experimental details for the DSPECs described in the main text in the framework of organic reactions.

Sensitizer	Catalyst	Reaction	Photocurrent	Details	Ref
*nanoITO*/TiO_2_@**4**	**7**	BZ	200 µA cm^−2^ (0.2 V vs. NHE)	Cronoamperometry in 20 mM acetate/acetic acid buffer pH 4.5/0.1 M LiClO_4_/0.1 M Benzyl alcohol	[97]
ITO(NP)@**38** (Assembly-TEMPO catalyst)		BZ	180 µA cm^−2^ (>0.4 V vs. SCE)	JV in 50 mM MeO-BA/carbonate buffer pH 10	[176]
TiO_2_@**39** (Assembly-TEMPO catalyst)		BZ	200 µA cm^−2^ (>0.4 V vs. SCE)	JV in 0.1 M borate buffer/50 mM [MeO-Ph-CH_2_OH]/0.1 M [NaClO_4_] pH 8. FE% = 80%	[28]
TiO_2_@**41**	TEMPO	Glycerol	400 µA cm^−2^ (0.1 V vs. Ag/AgCl)	Cronoamperometry in 1.0 M TEMPO 3 mm redox-gel (10% wt. PVDF-HFP, 1.2 M LiTFSI in ACN)/0.1 M glycerol aqueous solution (sat. NaCl, NaHCO_3_ pH 8.3. Stability >20 h	[182]
TiO_2_ NRAs@**1**	ACT	Lignin	690 μA cm^−2^ (0.1 V vs. Ag/AgCl)	Chopped cronoamperometry in 3 mM ACT/1.5 mM LMC/ACN	[185]

## Data Availability

No datasets were generated or analyzed during the current study.

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
