# Peer review of "Metal Complexes for Dye-Sensitized Photoelectrochemical Cells (DSPECs)"

_molecules, 2024, doi:10.3390/molecules29020293_

Round 1
Reviewer 1 Report
Comments and Suggestions for Authors
The review effectively traces the evolution of Dye-Sensitized Photoelectrochemical Cells (DSPECs) from Mallouk's seminal work, highlighting advancements in stabilization strategies. It is considered to have value as a review article. However, it is essential for a review paper to include the latest developments and provide the reader with the most up-to-date information.
It is suggested to further strengthen the presentation of experimental data, consider the stability of materials and environmental factors, improve the clarity of the paper structure, and conduct more theoretical analysis.
Reviewer 2 Report
Comments and Suggestions for Authors
The manuscript reviewed the current technology of Dye-Sensitized Photoelectrochemical Cells (DSPECs) and mainly focused on metal complexes outlining stabilization strategies and applications, which I found to be interesting. After reading the article carefully, we have concluded that the article has shown effort. However, it still needs a few changes before it is ready to be published.
1. First and foremost, the manuscript lacks summarized important parameters from the previous articles into a simple form. It is strongly recommended to include a comparison table, showing all the significant output parameters such as PCE, type of sensitizer, catalysts and other prominent elements to compare the effect on the cell performance.
2. The advancement in Dye-Sensitized Photoelectrochemical Cells (DSPECs) with time and cell efficiency can be drawn in a Figure to demonstrate the right path of the current development. Please provide such a conclusion of the figure accordingly.
3. Please enlarge the dimension and font size in Figures 32 and 33(b). The font size in the figures should be an almost similar size to the text.
Reviewer 3 Report
Comments and Suggestions for Authors
This manuscript comprehensively reviews the application of metal complexes in dye sensitized photoelectrochemical cells. It focuses on the stabilization strategies. It also summarizes some other oxidative reactions like HX and organics oxidation. The manuscript is well-written and logically structured. I have only some minor comments to be addressed before considering it for publication:
1. Authors are advised to compare in a table the photoelectric performance of the state-of-the-art photocatalysts and sensitizers with respect to conversion efficiency, applied potential, etc. This would make it easily accessible for readers.
2. The discussion on semiconductors and sensitizers can consider other relevant works e.g. Int. J. Electrochem. Sci., 6 (2011) 3316 – 3332, ACS Appl. Mater. Interfaces 2023, 15, 29, 35251–35260, Int. J. Electrochem. Sci., 7 (2012) 3610-3626, Coord. Chem. Rev. 248 (2004) 1343
3. One can hint to other semiconductor materials in the introduction such as ZnO e.g. Antioxidants 2023, 12(6), 1201, 10.3390/app12073442
